# Comparison of the mechanism of antimicrobial action of the gold(I) compound auranofin in Gram-positive and Gram-negative bacteria

Laísa Quadros Barsé,[1] Agnes Ulfig,[1] Marharyta Varatnitskaya,[1] Melissa Vázquez-Hernández,[2] Jihyun Yoo,[1] Astrid M. Imann,[1,3] Natalie Lupilov,[1] Marina Fischer,[4] Katja Becker,[4] Julia E. Bandow,[2] Lars I. Leichert[1]

**ABSTRACT**   While highly effective at killing Gram-positive bacteria, auranofin lacks significant activity against Gram-negative species for reasons that largely remain unclear. Here, we aimed to elucidate the molecular mechanisms underlying the low susceptibility of the Gram-negative model organism *Escherichia coli* to auranofin when compared to the Gram-positive model organism *Bacillus subtilis*. The proteome response of *E. coli* exposed to auranofin suggests a combination of inactivation of thiol-containing enzymes and the induction of systemic oxidative stress. Susceptibility tests in *E. coli* mutants lacking proteins upregulated upon auranofin treatment suggested that none of them are directly involved in *E. coli*'s high tolerance to auranofin. *E. coli* cells lacking the efflux pump component TolC were more sensitive to auranofin treatment, but not to an extent that would fully explain the observed difference in susceptibility of Gram-positive and Gram-negative organisms. We thus tested whether *E. coli*'s thioredoxin reductase (TrxB) is inherently less sensitive to auranofin than TrxB from *B. subtilis*, which was not the case. However, *E. coli* strains lacking the low-molecular-weight thiol glutathione, but not glutathione reductase, showed a high susceptibility to auranofin. Bacterial cells expressing the genetically encoded redox probe roGFP2 allowed us to observe the oxidation of cellular protein thiols *in situ*. Based on our findings, we hypothesize that auranofin leads to a global disturbance in the cellular thiol redox homeostasis in bacteria, but Gram-negative bacteria are inherently more resistant due to the presence of drug export systems and high cellular concentrations of glutathione.

**IMPORTANCE**   Auranofin is an FDA-approved drug for the treatment of rheumatoid arthritis. However, it has also high antibacterial activity, in particular against Gram-positive organisms. In the current antibiotics crisis, this would make it an ideal candidate for drug repurposing. However, its much lower activity against Gram-negative organisms prevents its broad-spectrum application. Here we show that, on the level of the presumed target, there is no difference in susceptibility between Gram-negative and Gram-positive species: thioredoxin reductases from both *Escherichia coli* and *Bacillus subtilis* are equally inhibited by auranofin. In both species, auranofin treatment leads to oxidative protein modification on a systemic level, as monitored by proteomics and the genetically encoded redox probe roGFP2. The single largest contributor to *E. coli*'s relative resistance to auranofin seems to be the low-molecular-weight thiol glutathione, which is absent in *B. subtilis* and other Gram-positive species.

**KEYWORDS**   auranofin, gold, drug repurposing, thioredoxin reductase, proteomics, *Escherichia coli*, *Bacillus subtilis*

Address correspondence to Lars I. Leichert, lars.leichert@ruhr-uni-bochum.de.

Laísa Quadros Barsé and Agnes Ulfig contributed equally to this article. Author order was determined alphabetically by last name.

The authors declare no conflict of interest.

See the funding table on p. 22.

10.1128/spectrum.00138-24   **1**

The evolutionary emergence and rapid spread of multidrug-resistant pathogenic bacteria is an ongoing and rising public health concern, as infections caused by these pathogens claim the lives of nearly 700,000 people across the globe each year (1). This number is expected to increase dramatically if no urgent actions are taken. Unfortunately, the slow progress in the discovery and development of new antibiotic classes has led to a limited availability of novel antimicrobial compounds for clinical use and, thus, made the effective treatment of many bacterial infections more challenging (2). Accordingly, there is an urgent medical need to identify new and improved antibacterial agents with unprecedented modes of action to tackle the global antibiotic resistance crisis (3). Since conventional *de novo* drug discovery and development are lengthy, expensive, and complex, repurposing approved drugs for antimicrobial therapy by re-evaluating their biological targets and mechanisms of action is considered a key strategy to overcome these issues (4). Although using known drugs for indications they were not originally developed for is far from new, it has gained considerable attention in the last decade, with about one-third of new US Food and Drug Administration (FDA) approvals being for repurposed drugs (5).

The use of metals, such as silver, gold, and platinum, and compounds based on these and other metals, in treating diseases dates to ancient times (6), but their full potential as therapeutic drugs has not been realized yet. With the discovery of the platinum-based compound cisplatin as an anticancer agent in the 20th century, metallodrugs have experienced an impressive renaissance of interest in modern medicine, leading to the development of several other metal complexes with antitumor and anti-inflammatory properties over the last decades (7).

Auranofin, a gold(I)-containing triethyl phosphine, was initially approved by the FDA in 1985 as an anti-inflammatory drug for the treatment of rheumatoid arthritis and was the first orally active gold compound proven to slow the progression of the disease by inhibiting antibody-mediated cellular toxicity and stimulating cell-mediated immunity (8, 9). Auranofin's primary mode of action in humans is thought to be the inhibition of thioredoxin reductase (TrxR) (10). Auranofin's exceptionally high affinity toward TrxR is due to the reactivity of its thiol ligand with thiol groups, especially the selenol group of the selenocysteine present in eukaryotic TrxR (11–14). TrxR was shown to be highly expressed in the inflamed joints of patients with rheumatoid arthritis, and it is assumed to be involved in disease activity and suppression of apoptosis in the affected tissue (15, 16). TrxR is a central component of the antioxidant thioredoxin (Trx) system. It catalyzes the NADPH-dependent reduction of the redox protein thioredoxin (Trx), which is crucial in maintaining the cellular environment in a reduced state and protecting the cells against damage from oxidative stress (17, 18).

More recent studies also indicate that TrxR is highly abundant in various types of human cancer cells and, thus, represents a promising therapeutic target for anticancer drug development. Gold-induced inactivation of mammalian TrxR by auranofin leads to severe oxidative stress due to the overproduction of reactive oxygen species (ROS) and activation of apoptotic signaling pathways, ultimately triggering cancer cell death (19, 20).

Besides these effects in eukaryotes, auranofin and related gold-based compounds have also shown promising activity against a wide range of pathogenic microorganisms at a clinically achievable nanomolar range. In a study by Debnath et al., auranofin was highly efficacious against the intestinal parasite *Entamoeba histolytica,* providing the first evidence that this gold drug could be repurposed as a novel antimicrobial agent (21). Indeed, more recent studies have reported remarkable bactericidal effects of auranofin against various pathogenic Gram-positive bacteria *in vitro* and *in vivo*, including *Mycobacterium tuberculosis*, certain *Bacillus* spp., *Enterococcus faecalis,* and methicillin-resistant *Staphylococcus aureus* (MRSA) (22–25). As a result, auranofin has become the reference compound for developing new, structurally related organogold antimicrobial drugs with lower toxicity, a broader spectrum of activity, and greater efficacy (26).

Although progress has been made in understanding the underlying mechanistical aspects of auranofin's antimicrobial activity, the precise mechanisms of its action and the respective biological targets in bacteria are still largely unknown.

Recent studies point toward a complex multitarget mechanism of auranofin's action in prokaryotic organisms that involves inhibition of several biosynthetic pathways, including protein, DNA, and cell wall synthesis (27), and disruption of thiol-redox homeostasis (22, 28). Among these possible modes of action, inhibition of thiol-containing redox enzymes essential for maintaining intracellular thiol-redox balance is widely considered the primary mechanism through which auranofin exerts its cytotoxic effects on bacteria (22). A growing number of studies demonstrate a strong inhibitory effect on bacterial thioredoxin reductase (TrxB). For instance, auranofin has been shown to effectively inhibit TrxB of *S. aureus*, *M. tuberculosis,* and *H. pylori in vitro*, with $IC_{50}$ values in the low nanomolar range. Notably, the growth of these organisms was also strongly impaired in the presence of auranofin (22, 29).

While auranofin is highly effective against Gram-positive bacteria, it is much less active against Gram-negative bacteria, exhibiting a markedly higher minimal inhibitory concentration (MIC) (22). However, the reasons for the limited susceptibility of Gram-negative bacteria to auranofin are still unknown.

In this work, we aimed to elucidate the molecular mechanisms underlying the increased tolerance of Gram-negative bacteria to auranofin. We utilized *E. coli* and *B. subtilis* as representative species of Gram-negative and Gram-positive bacteria, respectively, to systematically uncover the mode of auranofin's action by testing its effect on TrxB activity and growth of these organisms. We studied the proteomic response of *E. coli* exposed to auranofin stress to uncover potential defense mechanisms that could mediate their increased resistance to auranofin. We also focused on factors that differ between Gram-positive and Gram-negative species, such as TolC, the outer membrane factor of an efflux pump in *E. coli*. Cells lacking TolC were more susceptible to auranofin, however not to an extent that would make TolC-dependent drug efflux the only factor explaining *E. coli*'s relative resistance to auranofin. A systematic evaluation of mutants lacking components of the thioredoxin system and the glutathione-dependent antioxidant system suggests that the presence of a high concentration of the low-molecular-weight thiol glutathione is a major factor contributing to *E. coli*'s tolerance. This is in line with cytoplasmic thiol oxidation caused by auranofin *in vivo*, which we observed using *E. coli* and *B. subtilis* cells expressing the well-known redox probe roGFP2. Overall, our data suggest that auranofin's mode of action against Gram-negative species does not significantly differ from that against Gram-positive species. Instead, it seems that the higher resistance of *E. coli* is caused in part by a combination of drug efflux and, more importantly, the presence of high concentrations of the low-molecular-weight thiol glutathione, which attenuate the cytotoxic effects of auranofin.

## MATERIALS AND METHODS

### Determination of the MIC of auranofin against Gram-negative and Gram-positive bacteria

Auranofin (Santa Cruz Biotechnology, Heidelberg, Germany) was tested for bactericidal activity against *Acinetobacter baumannii* DSM 30007, *Pseudomonas aeruginosa* DSM 50071, *Bacillus subtilis* 168, *Staphylococcus aureus* DSM 20231, *Staphylococcus aureus* ATCC 43300 (MRSA), *E. coli* DSM 30083, *E. coli* BW25113, and *E. coli* BW25113-derived mutant strains from the KEIO collection (30) listed in Table 1. *E. coli* DSM 30083, *A. baumannii*, *S. aureus*, and *B. subtilis* were grown in Mueller Hinton broth, and *P. aeruginosa* in cation-adjusted Mueller Hinton II. MIC assays with these strains were performed in a microdilution assay in 200 µL medium according to CLSI guidelines, as described by Albada et al. (31). For MIC assays with *E. coli* BW25113 and the derived mutant strains from the KEIO collection, cells inoculated to a density of $5 \times 10^5$ cells/mL were grown at 37°C in reaction tubes containing 5 mL MOPS Minimal Medium (Teknova) (32), a

**TABLE 1** Bacterial strains and plasmids used in this study[a]

| | Relevant properties | Source/reference |
|---|---|---|
| *Bacterial strains* | | |
| *A. baumannii* DSM 30007 | Type strain | DSMZ (German collection of microorganisms and cell culture; Braunschweig, Germany). |
| *B. subtilis* 168 | Wild-type strain | DSMZ, strain DSM402 |
| *B. subtilis* LQB001 | *B. subtilis* 168 containing chromosomally inserted roGFP2 under the control of a xylose-inducible promotor in the *amyE* locus | This work. |
| *E. coli* DH5α | Cloning strain | New England Biolabs, Ipswich, MA, USA |
| *E. coli* BL21 (DE3) | Strain for heterologous protein expression from pET vectors | Stratagene, Santa Clara, CA. |
| *E. coli* DSM 30083 | Type strain | DSMZ |
| *E. coli* MG1655 | Wild-type strain | ATCC Manassas, VA, US, strain ATCC 700926 |
| *E. coli* BW25113 | Parental strain Keio collection (30) | The National BioResource Project (National Institute of Genomics, Japan) |
| *E. coli* JW5503 | *E. coli* BW25113 Δ*tolC* | The National BioResource Project (National Institute of Genomics, Japan) |
| *E. coli* JW3467 | *E. coli* BW25113 Δ*gor* | The National BioResource Project (National Institute of Genomics, Japan) |
| *E. coli* JW2663 | *E. coli* BW25113 Δ*gshA* | The National BioResource Project (National Institute of Genomics, Japan) |
| *E. coli* JW5156 | *E. coli* BW25113 Δ*trxA* | The National BioResource Project (National Institute of Genomics, Japan) |
| *E. coli* JW0871 | *E. coli* BW25113 Δ*trxB* | The National BioResource Project (National Institute of Genomics, Japan) |
| *E. coli* AI001 | DH5α pAI (*E. coli* trxB in pET22b(+)) | This work |
| *E. coli* AI002 | BL21 (DE3) pAI (*E. coli* trxB in pET22b(+)) | This work |
| *E. coli* LQB001 | BW25113 pCC_roGFP2 | This work. |
| *E. coli* MM003 | DH5α pMM (*B. subtilis* trxB in pET22b(+)) | This work |
| *E. coli* MM004 | BL21 (DE3) pMM (*B. subtilis* trxB in pET22b(+)) | This work |
| *P. aeruginosa* DSM 50071 | Type strain | DSMZ (German collection of microorganisms and cell culture; Braunschweig, Germany). |
| *S. aureus* DSM 20231 | Type strain | DSM (German collection of microorganisms and cell culture; Braunschweig, Germany). |
| *S. aureus* ATCC 43300 (MRSA) | Methicillin-resistant | ATCC, Manassas, VA, US. |
| Plasmids | | |
| pET22b(+) | IPTG-inducible plasmid (high copy number) | Novagen |
| pMM | pET22b(+) encoding *B. subtilis* 168 TrxR with an N-terminal SUMO tag | This study |
| pAI | pET22b(+) encoding *E. coli* MG1655 TrxR with an N-terminal SUMO tag | This study |
| pE-SUMO | SUMO tag | Lab collection |
| pCC_roGFP2 | pCC encoding roGFP2. IPTG-inducible plasmid, amp$^R$ | (33) |
| pSG1729_roGFP2 | pSG1729 (34) encodes roGFP2 under a xylose-inducible promotor for chromosomal insertion into the amyE locus of *B. subtilis*. amp$^R$, spec$^R$ | This work |
| pMX_roGFP2 | pMX encoding roGFP2 codon-optimized for *B. subtilis* | Thermo Fischer Scientific |

[a]Routine strain handling was performed on agar plates containing the appropriate antibiotics. All experiments were performed in compliance with German laws regarding biosafety and biosecurity.

medium which contains MOPS (3-(N-morpholino)propanesulfonic acid), supplemented with 10 µM thiamine and the respective concentrations of auranofin overnight.

Auranofin was dissolved in dimethylsulfoxide (DMSO) to a final concentration of 10 mg/mL. Serial dilutions of this stock solution were prepared in the various culture media with the Tecan Freedom Evo 75 liquid handling workstation (Tecan, Männedorf, Switzerland) for tests according to CLSI guidelines. They typically covered a range from 512 to 0.5 µg/mL (512, 256, 128, 64, 32, 16, 8, 4, 2, 1, and 0.5 µg/mL). In *E. coli* BW25113 and the *E. coli* BW25113-derived mutants, a range of 256–4 µg/mL (256, 128, 64, 32, 16, 8, and 4 µg/mL) was tested; these dilutions were prepared manually. Equivalent volumes of DMSO were used as vehicle controls but had no effect on bacterial growth overnight.

All bacterial strains were grown until the exponential growth phase was reached. At an $OD_{600}$ of 0.4–0.5, $5 \times 10^5$ cells/mL of the exponentially growing cultures were used to inoculate the previously prepared auranofin dilutions in the same growth medium. After 16–18 h of incubation at 37°C, bacterial growth at various auranofin concentrations was assessed visually and photometrically by measuring the cultures' OD at 600 nm. The lowest auranofin concentration with no visually perceptible growth was recorded as MIC.

## Construction of plasmids encoding bacterial TrxB

Plasmids and oligonucleotides used in this study are listed in Tables 1 and 2, respectively. Plasmid construction followed standard procedures. The correctness of all newly constructed plasmids was verified by DNA sequencing.

For the expression and subsequent purification of *E. coli* MG1655 and *B. subtilis* 168 TrxB, the respective *trxB* genes were fused to an N-terminal SUMO tag and cloned into pET22b(+) vectors (35). The SUMO tag contains a SUMO (Small Ubiquitin-like Modifier) protein sequence, which enhances expression and solubility of the fusion protein, a $His_6$ tag, which allows for purification of the fusion protein using Ni–NTA chromatography, and an Ulp1 protease cleavage site (35).

The SUMO tag sequence was amplified from the pE-SUMO plasmid using the forward primer AI003_SUMO_for carrying a NdeI restriction site and AI004_SUMO_rev or MM004_SUMO_rev, containing an overhang complementary to the N-terminal portion of the *E. coli* or *B. subtilis trxB* sequence, respectively (Table 2). In a second PCR, the genes encoding TrxB were amplified from genomic DNA of *E. coli* MG1655 and *B. subtilis* 168 using the following primers: AI001_for and AI002_rev, and MM001_for and MM002_rev for *E. coli* and *B. subtilis*, respectively. The resulting PCR fragments contained an overhang sequence to the SUMO tag at the 5′ end and an XhoI restriction site at the 3′ end. Finally, an overlap PCR was performed using the primers AI003_SUMO_for and AI002_rev (for *E. coli* TrxB) or MM002_rev (for *B. subtilis* TrxB) to fuse the SUMO tag sequence to the N-terminus of the respective *trxB* gene. The obtained PCR product (i.e., NdeI-SUMO-TrxB-XhoI) was then restricted with NdeI and XhoI and cloned into a pET22b(+) vector,

**TABLE 2** Oligonucleotides used in this study

| Oligonucleotide | Sequence (5′ → 3′) |
|---|---|
| AI001_for | GAACAGATTGGAGGTGGCACGACCAAAC |
| AI002_rev | GCGACTCGAGTTATTTTGCGTCAGCTAAAC |
| AI003_SUMO_for | GCGTCATATGGGTCATCACCATCATC |
| AI004_SUMO_rev | GGTCGTGCCACCTCCAATCTGTTC |
| MM001_for | AACAGATTGGAGGTGTGTCAGAAGAAAAAATTTAT GACGTG |
| MM002_rev | GCGCCTCGAGTTATTTTTAAGGTTTTCAGCGTTTCTTG GAGC |
| MM004_SUMO_rev | TTTTTCTTCTGACACACCTCCAATCTGTTCGCGGTGAG |
| MV001_for | CATCATATGAGCGATAAAATTATTCACC |
| MV002_rev | GAACTCGAGTTACGCCAGGTTAGCGTCGAGG |
| LQB001_for | AAACATATGATGGCTATCGTAAAAGCAA |
| LQB002_rev | AAAGGATCCTTAAAGATGTTTGTTTACAA |
| JY001_for | AAAGGTACCATGGTTAGCAAAGGC |
| JY002_rev | AAAAAGCTTTTACTTGTACAGTTCATCCATG |

resulting in pAI (*E. coli* MG1655 *trxB*) and pMM (*B. subtilis* 168 *trxB*). *E. coli* DH5α and *E. coli* BL21(DE3) cells were then transformed with the plasmids for plasmid maintenance and heterologous protein expression, respectively.

## Heterologous protein expression

For heterologous protein expression, a single colony of *E. coli* BL21 (DE3) carrying pAI or pMM was used to inoculate 50 mL LB broth containing 200 µg/L ampicillin, and the cells were grown overnight at 37°C and 125 rpm. The overnight culture was then used to inoculate 5 L of LB medium, supplemented with ampicillin. Cells were incubated at 37°C and 125 rpm until an $OD_{600}$ of 0.5–0.6 was reached. Subsequently, protein expression was induced by adding 1 mM isopropyl 1-thio-ß-D-galactopyranoside (IPTG) to the cell culture. After overnight incubation at 20°C and 125 rpm, cells were harvested by centrifugation at 7,800 × *g* and 4°C for 45 min, and pellets were stored at −80°C.

## Purification of ecTrxB and bsTrxB

*E. coli* BL21 (DE3) cell pellets obtained after expression of *B. subtilis* 168 (*bs*TrxB) or *E. coli* MG1655 TrxB (*ec*TrxB) and were resuspended in 30 mL lysis buffer 1 (50 mM sodium phosphate, 300 mM NaCl, pH 8) containing 2 mL of ethylenediaminetetraacetic acid (EDTA)-free protease inhibitor mixture (Roche Applied Science, Penzberg, Germany). Cells were disrupted by passing the cell suspension three times through a constant cell disruption system (TS benchtop; Constant Systems, Daventry, UK) at 1.9 kbar and 4°C, followed by the addition of the serine protease inhibitor phenylmethylsulfonyl fluoride (PMSF) to a final concentration of 1 mM. The cell lysate was centrifuged at 6,700 × *g* and 4°C for 1 h. The supernatant was vacuum-filtered through a 0.45-µm filter. Proteins were purified using Immobilized Metal Affinity Chromatography (IMAC). The supernatant was loaded onto a Pierce Centrifuge Column pre-packed with 2 mL HisPur Ni-NTA-Resin (Thermo Fischer Scientific) and pre-equilibrated with lysis buffer 1. The column was then washed with lysis buffer 1, supplemented with 10 mM imidazole, followed by one washing step with lysis buffer 1 containing 20 mM imidazole. Proteins were eluted by adding elution buffer 1 (50 mM sodium phosphate, 300 mM NaCl, 250 mM imidazole, pH 8.0). Elution fractions containing the respective protein were combined, filled to 50 mL with lysis buffer 1, and concentrated to 1 mL using a Vivaspin 20 PES, MWCO 10000 concentrator system (Sartorius Stedim Biotech). The latter step was repeated twice to reduce imidazole concentration in the buffer. To remove the SUMO tag, *ec*TrxB and *bs*TrxB were digested with a $His_6$-tagged SUMO Ulp1 protease, using a 75-fold excess (wt/wt) protein to protease. After overnight incubation at 4°C, the reaction mixtures were purified using IMAC as described above. Wash fractions containing the purified and untagged TrxBs were concentrated using Vivaspin 20 PES, MWCO 10000 concentrator system (Sartorius Stedim Biotech) to a final volume of 1 mL. Proteins were stored in lysis buffer 1 supplemented with 10% (vol/vol) glycerol at −80°C.

## Analysis of TrxB inhibition by auranofin using a DTNB reduction assay

TrxB activity in the absence or presence of auranofin was determined by performing a 5,5′-dithiobis(2-nitrobenzoic) acid (DTNB) reduction assay. In this NADPH-dependent assay, TrxB reduces DTNB directly to yield the spectrophotometrically detectable 5-thio-2-nitrobenzoic acid (TNB). DTNB assays were performed in transparent, flat-bottom 96-well plates (Sarstedt, Nümbrecht, Germany) as described by Lu et al. (36). Briefly, 1 µM *ec*TrxB or *bs*TrxB was added to TE buffer (50 mM Tris-HCl, 2 mM EDTA, pH 7.5) containing 200 µM NADPH (Sigma-Aldrich, St. Louis, USA) for 5 min, in a total volume of 100 µL. The enzymatic reaction was then initiated by adding 100 µL TE buffer containing 2 mM DTNB (Sigma-Aldrich, St. Louis, USA). The amount of liberated TNB was then quantified spectrophotometrically by measuring the absorbance at 412 nm for 15 min. Experiments containing auranofin ranging from 0.1 to 2 µM (0.1, 0.2, 0.5, 1, 2 µM) were performed in triplicate. The remaining TrxB activity after adding auranofin was normalized to activity in the absence of auranofin, which was set to 100%.

## Proteomic profiling of auranofin-stressed *E. coli* cells using radioactive labeling and 2D-PAGE

2D PAGE-based proteomic analysis was performed as previously described (37). For the analysis of the proteomic response of *E. coli* to auranofin exposure, *E. coli* MG1655 was used. Briefly, a stock of auranofin was dissolved in DMSO to a final concentration of 10 mg/mL. *E. coli* MG1655 cells were grown in Neidhardt MOPS minimal medium (32) at 37°C and 200 rpm until an $OD_{500}$ of 0.35 was reached. Cultures were split into an untreated control and a culture treated with a sublethal concentration of 120 µg/mL. After 10 min, newly synthesized proteins were labeled with 0.37 MBq/mL L-[$^{35}$S]-methionine (Hartmann Analytic, Braunschweig, Germany). After 5 min, protein synthesis was stopped by the addition of 100 µg/mL chloramphenicol and non-radio-active L-methionine (to a final concentration of 1 mM L-methionine and 10 mM Tris originally set to pH 7.5 in the stock solution). Cells were harvested by centrifugation and washed twice with 100 mM Tris/1 mM EDTA, pH 7.5. The cells were disrupted by ultrasonication with a VialTweeter (Hielscher, Teltow, Germany), and the supernatant was recovered after centrifugation. The protein concentration was determined by a Bradford assay (Roti NanoQuant, Roth, Karlsruhe, Germany). Protein biosynthesis rates were determined by measuring the incorporation of radioactive methionine in auranofin-treated and untreated samples after protein precipitation with 20% (wt/vol) trichloro acetic acid using a scintillation counter (Tri-Carb 2800TR, PerkinElmer). 55 µg of protein from pulse-labeled samples for analytical gels and 300 µg of protein from unlabeled samples for preparative gels were rehydrated into IPG strips (24 cm, pH 4–7; GE Healthcare, Chalfont St Giles, UK). A Multiphor II (GE Healthcare) was used for isoelectric focusing, and for the second dimension PAGE, the Ettan DALTwelve system (GE Healthcare) was used. The gels were stained with ruthenium(II)tris(4,7-diphenyl-1,10-phenantrolin disulfonate) and washed with ethanol and acetic acid (38). Fluorescent gel images were taken with the Typhoon Trio$^+$ Variable Mode Imager (GE Healthcare) using an excitation wavelength of 532 nm and an emission filter of 600 nm. Radio-labeled analytical gels were dried onto Whatman paper and exposed to Storage Phospho Screens (GE Healthcare). The phospho screens were scanned with the Typhoon Trio$^+$ with an excitation wavelength of 633 nm and an emission filter of 390 nm. For visualization and analysis of the 2D gels, the Decodon Delta 2D image analysis software was used (39). Two biological replicates with one technical replicate each were analyzed against their respective controls for quantitation. Protein spots detected as twofold upregulated compared to the untreated control based on relative signal intensity were defined as marker proteins upon visual inspection (37).

Protein identification from gel spots was performed as described previously (40). Spots were excised from a non-radioactive 2D gel and destained with 20 mM ammonium bicarbonate in 30% acetonitrile. The protein spots were digested with either 12.5 ng/µL trypsin for low-intensity or 6.25 ng/µL trypsin for high-intensity spots (Promega, Fitchburg, FI, USA) for 18 h at 37°C. Peptides were eluted in 0.1% trifluoro-acetic acid in MS-grade water and ultrasonicated for 15 min once (or twice if they were low-intensity spots). For protein identification, the samples were analyzed using a nanoACQUITY-UPLC-coupled SYNAPT G2-S high-definition mass spectrometer equipped with a NanoLockSpray Source for electrospray ionization and a time-of-flight detector (Waters, Milford, USA). The samples were loaded into a trap column (C$_{18}$, pore size 100 Å, particle diameter of 5 µm and an inner diameter of 180 µm, length 20 mm) and eluted with 0.1% formic acid/acetonitrile gradient (350 µL/min, linear gradient 2%–5% acetonitrile in 2 min, 5%–60% acetonitrile in 20 min) from an analytical column (C$_{18}$, pore size 130 Å, particle diameter of 1.7 µm and an inner diameter of 75 µm, length 150 mm) at 40°C and subjected to mass spectrometry. The spectra were recorded in a mass range of 50–1,800 m/z with a 0.5 s/scan in positive mode. The following parameters were used for the NanoLockSpray source: capillary voltage of 1.5 kV; sampling cone voltage of 30 V; source temperature of 100°C; desolvation temperature of 150°C; cone glass flow 50 L/h; and desolvation gas flow of 550 L/h. Leucine enkephalin was used as a lock mass analyte

using the lock spray channel (lock spray capillary voltage, 2.5 kV). Analysis of spectra was performed by MassLynx V4.1 SCN813 (Waters).

For protein identification, a non-redundant version of the *E. coli* MG1655 protein database (NCBI Bio Project accession PRJNA647163) with 4740 protein entries (plus the sequences of trypsin and human keratin) and the ProteinLynxGlobalServer (PLGS) software V3.5.3 (Waters) with the following parameters were used: chromatographic peak width, automatic; MS ToF resolution, automatic; lock mass window, 0.3 Da; low energy threshold, 50 counts. Other parameters were as follows: peptide tolerance, automatic; fragment tolerance, automatic; minimal fragment ion matches per peptide, 2; minimal fragment ion matches per protein, 5; minimal peptide matches per protein, 2; maximum protein mass, 250 kDa; primary digest reagent, trypsin; secondary digest reagent, none; missed cleavages, 1; fixed modifications, carbamidomethyl C; variable modifications, deamination N and Q, oxidation M; false-positive rate, 4%. The following parameters were considered for protein assignment: Coverage of identified protein was >10%, at least three peptide ions and five product ions were assigned, and theoretical pI and MW were consistent with the spot's position on the 2D gel.

## RoGFP2 expression in bacteria

To measure the redox state in the cytoplasm of *E. coli* MG1655 and *B. subtilis* 168, cells were transformed with the pCC_roGFP2 plasmid as described (33) and pSG1729_roGFP2, respectively. pSG1729_roGFP2 cannot replicate in *B. subtilis*, but part of it integrates into the *amyE* locus on the chromosome through a double-crossover event. To create pSG1729_roGFP2, the EGFP insert from pSG1729 (34) was removed using KpnI and HindIII restriction enzymes. The gene sequence of roGFP2, codon-optimized for *B. subtilis*, was amplified from pMX_roGFP2 (Thermo Fischer Scientific) by PCR. Restriction sites for KpnI and HindIII were introduced using the primers JY001_for and JY001_rev. Plasmid construction followed standard procedures and was performed in *E. coli* DH5α. After successful plasmid construction, confirmed by sequencing, pSG1729_roGFP2 was transformed into *B. subtilis* 168 using standard protocols (Anagnostopoulos and Spizizen, 1961), and selection for chromosomal insertion of the plasmid was performed on LB plates containing 100 µg/mL spectinomycin. Double crossover of positive clones was confirmed by the inability to hydrolyze starch on LB plates containing 0.5% starch and subsequent PCR amplification of the chromosomal insert and sequencing of the PCR product.

For roGFP2 expression in *E. coli* MG1655, cells were grown in MOPS medium at 37°C and 125 rpm. 200 µM of IPTG was used to induce protein expression at an $OD_{600}$ of 0.4–0.6, and roGFP2 was subsequently expressed for 16 h at 20°C and 125 rpm in an MOPS medium. After expression, cells were harvested, washed in 4-(2-hydroxyethyl)-1-piperazineethanesulfonic acid (HEPES) buffer (40 mM, pH 7.4), and adjusted to an $OD_{600}$ of 1.2.

For roGFP2 expression in *B. subtilis* 168, a single colony was inoculated in DSM medium [Bacto nutrient broth 0.8% (wt/vol), KCl 13.4 mM, $MgSO_4$ 1 mM, $Ca(NO_3)_2$ 1 mM, $MnCl_2$ 10 µM, $FeSO_4$ 1 µM] containing 0.1% xylose (wt/vol), and cells were grown for 16 h at 30°C and 200 rpm. Cells were harvested, washed in BMM medium (41), and incubated for 30 min at 25°C and 800 rpm in a ThermoMixer (Eppendorf, Hamburg, Germany). The final $OD_{600}$ was adjusted to 5. Cells were treated with 200 µM aldrithriol-2 (Sigma-Aldrich) (AT-2) as an oxidation control and 2 mM dithiothreitol (Sigma-Aldrich) as a reduction control. 0.5, 1, and 5 µM auranofin were used as treatment. As AT-2 and auranofin were dissolved in DMSO, an equivalent volume of DSMO was added to all samples as vehicle control. 1 mL of cell suspension was added to a 1,500 µL Quartz Suprasil fluorescence cuvette (Hellma, Müllheim, Germany) with a stirring bar. Measurements were done at 25°C in a JASCO FP-8500 fluorescence spectrometer equipped with the temperature-controlled sample holder "EHC-813" at 25°C for 60 min under continuous stirring. Measurement parameters in *E. coli* were as follows: 510 nm (Em), 350–500 nm (Ex), 5 nm slit width (Ex/Em), and medium sensitivity. After stabilization, respective treatments were added at the indicated final concentrations, and a time

series of 61 additional spectra was recorded. Measurement parameters for *B. subtilis* were as follows: 520 nm (Em), 350–500 nm (Ex), 10 nm slit width (Ex/Em), and high sensitivity. After stabilization, respective treatments were added at the indicated final concentrations, and a time series of 61 additional spectra was recorded. The ratios of the fluorescence excitation intensities (405/488 nm) were used to calculate the probe's oxidation state. All values were normalized to fully oxidized (AT-2-treated) and fully reduced (DTT-treated) roGFP2 with the following equation (1):

$$OxD = \frac{R - R_{red}}{\left(\frac{I_{488}ox}{I_{488}red}\right) * (R_{ox} - R) + (R - R_{red})}$$

$R_{ox}$ is the 405/488 nm ratio of oxidized (AT-2 treated) and $R_{red}$ of reduced (DTT-treated) cells expressing roGFP2, respectively. $I_{488}ox$ and $I_{488}red$ are the fluorescence intensities at 488 nm under oxidizing (i.e., AT-2) or reducing (i.e., DTT) conditions. $R$ is the measured 405/488 nm ratio of the sample.

For calculation and normalization of the probe's oxidation degree at each time point, the means of all values recorded at all time points for $R_{red}$, $R_{ox}$, $I_{488}ox$, and $I_{488}red$ were used. Data were processed using Microsoft Excel software and GraphPad Prism version 10.1.0.

## RESULTS

### Auranofin is antibacterial but less effective against *E. coli* and other Gram-negative species

Previous studies have shown that auranofin possesses potent bactericidal activity against many Gram-positive bacteria but seems to be markedly less effective against pathogenic Gram-negative species (22–24). However, the reasons for the poor susceptibility of Gram-negative bacteria to auranofin are not entirely clear. We analyzed the antimicrobial activity of auranofin against Gram-negative and Gram-positive bacterial strains by MIC assays. For this purpose, we added various concentrations of auranofin to a defined number of bacterial cells of *B. subtilis* 168, *S. aureus* DSM 20231/ATCC 43300, *P. aeruginosa* DSM 50071, *A. baumanni* DSM 30007, and *E. coli* DSM 30083 and examined their growth after overnight incubation. The MIC was defined as the lowest auranofin concentration, at which no visibly perceptible bacterial growth was observed. In line with expectations, auranofin showed potent growth inhibition against Gram-positive bacteria, with an MIC of 0.5 µg/mL for each Gram-positive strain tested (Table 3). The MIC values determined for auranofin against *S. aureus* and *B. subtilis* correlate with those reported in previous studies (22–24). By contrast, auranofin exhibited poor antimicrobial activity against Gram-negative bacteria, with MICs of 128 µg/mL, 32 µg/mL, and 512 µg/mL against *E. coli*, *A. baumannii,* and *P. aeruginosa*, respectively, which agrees with previous reports (22, 27).

TABLE 3   MICs of auranofin against various Gram-negative and Gram-positive bacterial species

| Strain | MIC, µg/mL |
|---|---|
| *Gram-negative bacteria* | |
| *E. coli* DSM 30083 | 128 |
| *A. baumannii* DSM 30007 | 32 |
| *P. aeruginosa* DSM 50071 | 512 |
| *Gram-positive bacteria* | |
| *B. subtilis* 168 | 0.5 |
| *S. aureus* DSM 20231 | 0.5 |
| *S. aureus* ATCC 43300 (MRSA) | 0.5 |

## Auranofin causes upregulation of proteins involved in central metabolism and the defense against oxidative stress in *E. coli*

To understand whether auranofin exposure leads to a markedly different cellular response in Gram-negative bacteria, we performed proteomics experiments in *E. coli*. We compared our results to the results previously obtained for *B. subtilis* (42). In our experiments, we treated an exponentially growing culture of *E. coli* MG1655 with a sublethal concentration of 120 µg/mL auranofin. This concentration was found to partially inhibit bacterial growth by 26.27% ± 4.14% (Fig. 1a). Upon exposure to auranofin for 10 min, the cells were pulse-labeled for 5 min with L-[$^{35}$S]-methionine, which was incorporated into newly synthesized proteins. Quantification of total L-[$^{35}$S]-methionine incorporation revealed no significant difference between the control and the auranofin-treated culture, suggesting that auranofin does not directly inhibit protein synthesis (Fig. 1b).

Newly synthesized proteins were quantified by densitometry of radioactive spots after two-dimensional PAGE. Auranofin-induced changes in protein synthesis patterns were visualized by overlaying the gel images of the L-[$^{35}$S]-methionine-labeled proteins under control conditions (green) and after treatment with auranofin (red). Protein spots synthesized only under control conditions appear in green; protein spots that were synthesized only upon auranofin treatment appear in red, whereas protein spots synthesized under both conditions appear yellow. Out of the three independent experiments, a representative dual-channel image produced from autoradiographs of 2D gels obtained for control and auranofin-treated cells is shown in Fig. 1c. Auranofin treatment caused a more than twofold increase in relative synthesis rates of 30 protein spots (indicated by arrows). From non-radioactive gels, proteins could be identified by mass spectrometry in 20 of those spots (Fig. 1c; Table 4).

Most identified marker proteins upregulated in response to auranofin treatment are associated with a specific response to oxidative stress and/or involved in cellular processes such as ROS detoxification, central metabolism, amino acid biosynthesis, and metal homeostasis.

Many of the upregulated proteins are enzymes of the central metabolic network, such as GpmM (phosphoglycerate mutase), PykF (pyruvate kinase I), Lpd (lipoamide dehydrogenase), and the alpha subunit of the ATP synthase F$_1$ complex of the electron transport chain AtpA. Auranofin treatment was also found to increase the synthesis of the enzyme dihydroorotase (PyrC), which functions in the pathway for the biosynthesis of pyrimidine nucleotides. Some other induced proteins, such as AhpF and AhpC, which are components of the alkyl hydroperoxide reductase, and Gor (glutathione oxidoreductase), belong to the OxyR regulon and are involved in the protection against peroxide and in maintaining redox balance (44, 45). Gor is responsible for recycling the glutathione pool NADPH-dependently. NADPH is thought to be delivered by the pentose pathway, of which RpiA was upregulated. A similar observation was made in our laboratory when *E. coli* cells were subjected to diamide or the antimicrobial compound allicin, both known to induce disulfide stress (46, 47): here, central metabolic enzymes and OxyR-dependent proteins were upregulated, as well. In addition, auranofin treatment led to the upregulation of a sulfur assimilation enzyme (CysN) for the biosynthesis of sulfur-containing amino acids and the oxidative stress-protective multicopper oxidase CueO.

To test whether one of these upregulated proteins is directly involved in *E. coli*'s resistance against auranofin, we tested, whether selected single mutants from the KEIO collection lacking these proteins were more susceptible to auranofin in an MIC assay. To exclude the interference of antioxidative media components with auranofin, we performed these tests in the chemically defined MOPS medium, the same medium used for the proteomic L-[$^{35}$S]-methionine labeling. The mutants lacking GpmM, Pgk, Lpd, and PyrC did not grow in MOPS medium. None of the proteins missing in the selected mutants that did grow, impacted the MIC when absent, except for NemA (Table 4). The

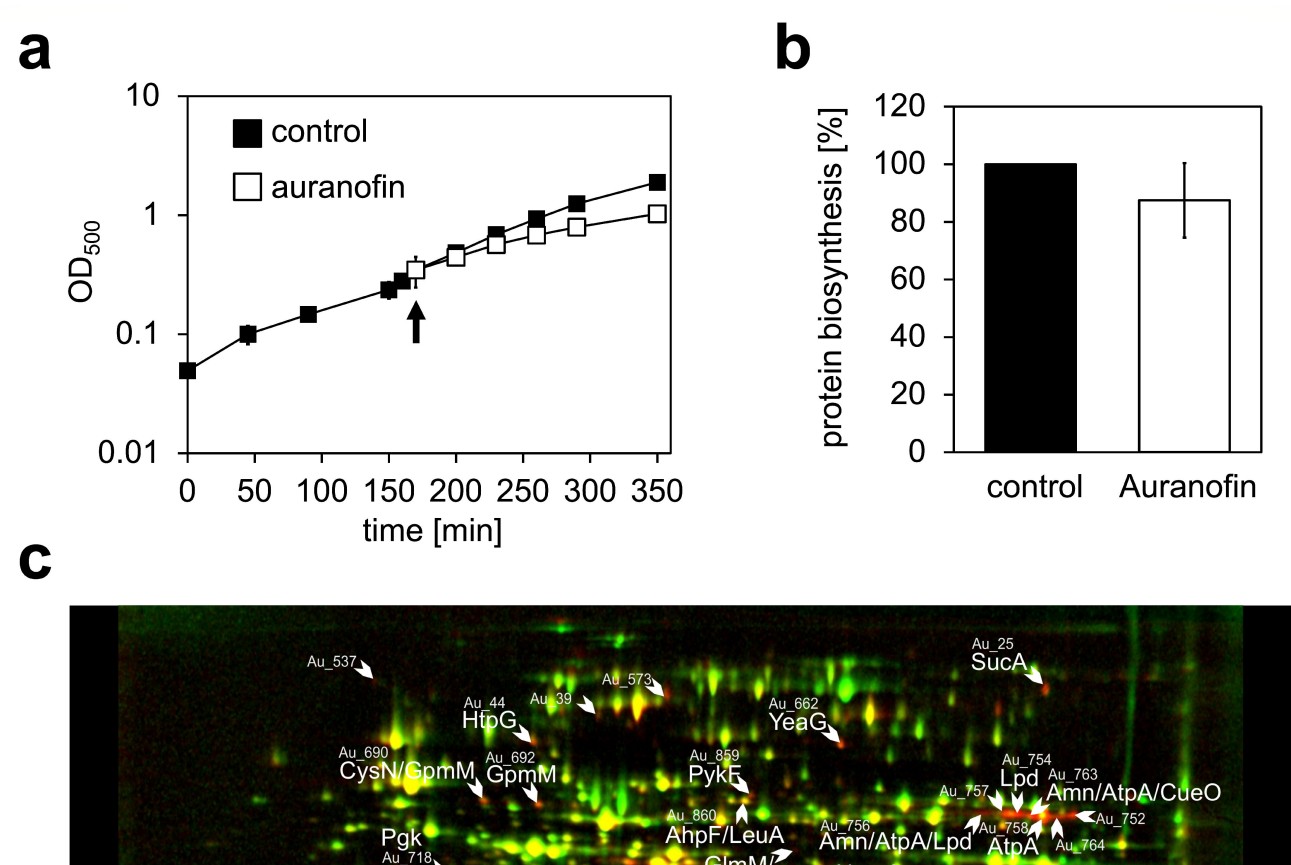

**FIG 1** Proteomic response of *E. coli* MG1655 stressed with 120 µg/mL auranofin. (a) Growth curves of *E. coli* MG1655 grown in MOPS media at 37°C and 200 rpm without auranofin (■) and after the addition of 120 µg/mL auranofin (□). An arrow indicates the timepoint of auranofin addition. (b) Incorporation of L-[³⁵S]-methionine during a 5-min pulse starting 10 min after treatment with 120 µg/mL auranofin. (c) 2D-PAGE analysis of the pulse-labeled proteomics response profile of *E. coli* MG1655 treated with 120 µg/mL auranofin. False-color overlay of treated and untreated radioactive gels shows proteins synthesized after pulse labeling for 5 min after treatment with auranofin (red, Au) or in the untreated control (green, Co). Upregulated proteins with a regulation factor ≥2

**Fig 1 (Continued)**

were considered marker proteins and were labeled by arrows and the Spot ID Au_X. Marker proteins identified by mass spectrometry are additionally labeled with the protein's name. For Spot ID, identified proteins, and functions, see Tables 4 and 5. (a and c) A representative example from three (a) or two (c) biological replicates. (b) The average and standard deviation of three biological replicates.

Δ*nemA* strain was found to be slightly more resistant to auranofin treatment and had an MIC of 256 µg/mL (WT being 128 µg/mL).

Overall, the proteomic results indicate that auranofin-treated cells suffer systemic oxidative stress. This could lead to the direct inactivation of essential thiol-containing enzymes, for example, in the central metabolism, and a depletion of free cysteine levels or, more generally, thiol levels. We made a similar observation in *B. subtilis* (42), and based on our proteome data, we could not differentiate between the mode of action of auranofin in Gram-positive and Gram-negative bacteria.

## The higher resistance against auranofin can be partially explained by more effective efflux systems in Gram-negative bacteria

Efflux pumps are critical players in antimicrobial resistance in Gram-negative bacteria. These pumps actively transport a variety of substrates, including antibiotics and heavy metals, out of the bacterial cell, thereby reducing the intracellular concentration of these

**TABLE 4** Identified marker proteins of auranofin-treated *E. coli* MG1655[a]

| Spot ID | Reg. factor | Protein (EcoCyc ID) | Function | MIC µg/mL |
|---|---|---|---|---|
| 497 | 2.3 | AhpC (EG11384) | Alkyl hydroperoxide reductase, AhpC component | 128 |
| 860 | 3.6 | AhpF (EG11385) | Alkyl hydroperoxide reductase, AhpF component | 128 |
| | | LeuA (EG11226) | 2-isopropylmalate synthase | n.d. |
| 763 | 4.6 | Amn (EG10039) | AMP nucleosidase | 128 |
| | | AtpA (EG10098) | ATP synthase F1 complex subunit α | 128 |
| | | CueO (EG12318) | Multicopper oxidase CueO | 128 |
| 756 | 4.0 | Amn (EG10039) | AMP nucleosidase | 128 |
| | | AtpA (EG10098) | ATP synthase F1 complex subunit α | 128 |
| | | Lpd (EG10543) | Lipoamide dehydrogenase | n.v. |
| 758 | 15.0 | AtpA (EG10098) | ATP synthase F1 complex subunit α | 128 |
| 347 | 3.2 | Bfr (EG10113) | Bacterioferritin | 128 |
| 690 | 2.1 | CysN (EG10194) | Sulfate adenylyltransferase subunit 1 | 128 |
| | | GpmM (EG12296) | 2,3-bisphosphoglycerate-independent phosphoglycerate mutase | n.v. |
| 782 | 4.0 | GlmM (EG11553) | Phosphoglucosamine mutase | n.d. |
| | | Gor (EG10412) | Glutathione reductase | 128 |
| | | Mpl (EG12440) | UDP-N-acetylmuramate-L-alanyl-γ-D-glutamyl-meso-2,6-diaminoheptanedioate ligase | n.d. |
| | | PhoA (EG10727) | Alkaline phosphatase | n.d. |
| 692 | 2.6 | GpmM (EG12296) | 2,3-bisphosphoglycerate-independent phosphoglycerate mutase | n.v |
| 44 | 4.1 | HtpG (EG10461) | Chaperone protein HtpG | n.d. |
| 754 | 7.2 | Lpd (EG10543) | Lipoamide dehydrogenase | n.v. |
| 620 | 2.7 | MglB (EG10593) | D-galactose/methyl-galactoside ABC transporter periplasmic binding protein | 128 |
| 328 | 2.4 | MoaB (EG11596) | Protein MoaB | 128 |
| 806 | 3.9 | NemA (G6890) | N-ethylmaleimide reductase | 256 |
| 718 | 6.1 | Pgk (EG10703) | Phosphoglycerate kinase | n.v. |
| 859 | 3.6 | PykF (EG10804) | Pyruvate kinase 1 | 128 |
| 816 | 3.0 | PyrC (EG10806) | Dihydroorotase | n.v. |
| | | YqhD (G7564) | NADPH-dependent aldehyde reductase YqhD | n.d. |
| 239 | 2.0 | RpiA (EG11443) | Ribose-5-phosphate isomerase A | n.d. |
| 25 | 3.2 | SucA (EG10979) | 2-oxoglutarate decarboxylase, thiamine-requiring | 128 |
| 662 | 3.9 | YeaG (G6969) | Protein kinase YeaG | 128 |

[a]Protein functions and accession number (EcoCyc ID) were taken from the EcoCyc database (43). Regulation factor (Reg. factor) is the average of 2 biological replicates. The MIC of the corresponding deletion mutant is shown (*n* = 3). Deletion mutants not viable in minimal medium are marked by n.v., MIC not determined is marked by n.d.

**TABLE 5** List of marker protein spots induced after treatment with auranofin in *E. coli* MG1655[a]

| Spot ID | Regulation factor ± range | Protein | EcoCyc accession number | Function | Theoretical MW [Da] | Theoretical pI | Peptides | Fragments | Coverage % |
|---|---|---|---|---|---|---|---|---|---|
| 25 | 3.2 ± 2.7 | **SucA** | EG10979 | 2-oxoglutarate decarboxylase, thiamine-requiring | 104,995 | 6.0 | 15 | 81 | 18.3 |
| 44 | 4.1 ± 0.7 | **HtpG** | EG10461 | Chaperone protein HtpG | 71,378 | 4.9 | 39 | 296 | 42.0 |
| | | CsiR (excl.) | EG12386 | DNA-binding transcriptional repressor GlaR | 24,974 | 6.2 | 5 | 15 | 18.2 |
| | | YciH (excl.) | EG11128 | Putative translation factor | 11,389 | 9.8 | 4 | 13 | 26.9 |
| | | AroK (excl.) | EG10081 | Shikimate kinase 1 | 19,526 | 5.1 | 2 | 5 | 17.3 |
| 239 | 2.0 ± 0.2 | **RpiA** | EG11443 | Ribose-5-phosphate isomerase A | 22,845 | 5.0 | 4 | 21 | 17.4 |
| 326 | 2.4 ± 0.3 | EptA (excl.) | EG11613 | Phosphoethanolamine transferase EptA | 61,627 | 6.4 | 2 | 9 | 9.5 |
| 328 | 2.4 ± 0.3 | **MoaB** | EG11596 | Protein MoaB | 18,653 | 5.7 | 6 | 83 | 25.3 |
| 347 | 3.2 ± 0.8 | **Bfr** | EG10113 | Bacterioferritin | 18,483 | 4.5 | 3 | 18 | 11.4 |
| 357 | 2.3 ± 0.1 | N/A | N/A | | | | | | |
| 374 | 2.3 ± 0.1 | FtsK (excl.) | G6464 | Cell division DNA translocase FtsK | 146,571 | 4.7 | 5 | 25 | 2.5 |
| 497 | 2.3 ± 0.4 | **AhpC** | EG11384 | Alkyl hydroperoxide reductase, AhpC component | 20,748 | 4.8 | 5 | 32 | 21.9 |
| 537 | 3.7 ± 0.2 | N/A | N/A | | | | | | |
| 606 | 4.4 ± 1.8 | YbcJ (excl.) | EG12879 | Putative RNA-binding protein YbcJ | 7,384 | 8.3 | 3 | 19 | 4.3 |
| | | XapR (excl.) | EG11146 | DNA-binding transcriptional activator XapR | 33,605 | 9.1 | 2 | 7 | 12.2 |
| 620 | 2.7 ± 0.7 | **MglB** | | D-galactose/methyl-galactoside ABC transporter periplasmic binding protein | 35,690 | 5.6 | 23 | 156 | 50.3 |
| 638 | 2.7 ± 0.7 | N/A | | | | | | | |
| 662 | 3.9 ± 2.5 | **YeaG** | G6969 | Protein kinase YeaG | 74,433 | 5.5 | 25 | 160 | 32.3 |
| 690 | 2.1 ± 0.7 | **CysN** | EG10194 | Sulfate adenylyltransferase subunit 1 | 52,526 | 4.8 | 45 | 336 | 47.8 |
| | | **GpmM** | EG12296 | 2,3-bisphosphoglycerate-independent phosphoglycerate mutase | 56,158 | 5.0 | 8 | 35 | 13.6 |
| | | YedJ (excl.) | EG12710 | Putative HD superfamily phosphohydrolase YedJ | 25,888 | 5.9 | 4 | 14 | 17.3 |
| 692 | 2.6 ± 1.3 | **GpmM** | EG12296 | 2,3-bisphosphoglycerate-independent phosphoglycerate mutase | 56,158 | 5.0 | 25 | 168 | 21.6 |
| | | IhfA (excl.) | EG10440 | Integration host factor subunit α | 11,346 | 9.9 | 2 | 7 | 19.2 |
| 718 | 6.1 ± 2.5 | **Pgk** | EG10703 | Phosphoglycerate kinase | 41,092 | 4.9 | 7 | 41 | 17.6 |
| 752 | 7.0 ± 0.4 | CueO (excl.) | EG12318 | Multicopper oxidase CueO | 56,519 | 6.3 | 5 | 46 | 9.7 |

*(Continued on next page)*

**TABLE 5** List of marker protein spots induced after treatment with auranofin in *E. coli* MG1655[a] (Continued)

| Spot ID | Regulation factor ± range | Protein | EcoCyc accession number | Function | Theoretical MW [Da] | Theoretical pI | Peptides | Fragments | Coverage % |
|---|---|---|---|---|---|---|---|---|---|
| Php | | Php | G7731 | Putative hydrolase | 32,894 | 5.1 | 4 | 15 | 20.9 |
| 754 | 7.2 ± 2.4 | Lpd | EG10543 | Lipoamide dehydrogenase | 50,656 | 5.7 | 10 | 93 | 25.7 |
| | | MalI (excl.) | EG10557 | DNA-binding transcriptional repressor MalI | 36,602 | 6.5 | 7 | 20 | 28.1 |
| 756 | 4.0 ± 2.0 | Lpd | EG10543 | Lipoamide dehydrogenase | 50,656 | 5.7 | 10 | 64 | 28.5 |
| | | Amn | EG10039 | AMP nucleosidase | 53,961 | 5.9 | 11 | 49 | 19.8 |
| | | AtpA | EG10098 | ATP synthase F1 complex subunit α | 55,187 | 5.7 | 6 | 38 | 10.7 |
| 757 | 4.1 ± 1.9 | Tcyj | G7039 | Cystine ABC transporter periplasmic binding protein | 29,021 | 6.2 | 4 | 21 | 12.0 |
| | | MepM (excl.) | EG10013 | Peptidoglycan endopeptidase MepM | 49,027 | 9.8 | 6 | 20 | 13.9 |
| 758 | 15.0 ± 6.8 | AtpA | EG10098 | ATP synthase F1 complex subunit α | 55,187 | 5.7 | 12 | 74 | 18.7 |
| | | Amn (excl.) | EG10039 | AMP nucleosidase | 53,961 | 5.9 | 7 | 31 | 9.1 |
| 763 | 4.6 ± 2.2 | Amn | EG10039 | AMP nucleosidase | 53,961 | 5.9 | 8 | 39 | 14.9 |
| | | AtpA | EG10098 | ATP synthase F1 complex subunit α | 55,187 | 5.7 | 13 | 64 | 21.8 |
| | | CueO | EG12318 | Multicopper oxidase CueO | 56,519 | 6.3 | 3 | 25 | 11.0 |
| | | IbpB (excl.) | EG11535 | Small heat shock protein IbpB | 16,083 | 5.0 | 4 | 12 | 19.7 |
| 764 | 4.9 ± 1.0 | AtpA (excl.) | EG10098 | ATP synthase F1 complex subunit α | 55,187 | 5.7 | 6 | 25 | 9.1 |
| | | ArgA (excl.) | EG10063 | N-acetylglutamate synthase | 49,164 | 6.1 | 5 | 20 | 8.5 |
| | | PyrC (excl.) | EG10806 | Dihydroorotase | 38,802 | 5.8 | 3 | 14 | 8.6 |
| 782 | 4.0 ± 0.6 | Gor | EG10412 | Glutathione reductase | 48,741 | 5.6 | 16 | 91 | 35.1 |
| | | Mpl | EG12440 | UDP-N-acetylmuramate—L-alanyl-γ-D-glutamyl-meso-2,6-diaminoheptanedioate ligase | 49,842 | 5.5 | 13 | 57 | 19.9 |
| | | GlmM | EG11553 | Phosphoglucosamine mutase | 47,513 | 5.6 | 9 | 50 | 20.0 |
| | | PhoA | EG10727 | Alkaline phosphatase | 49,408 | 5.7 | 11 | 45 | 31.2 |
| 806 | 3.9 ± 0.1 | NemA | G6890 | N-ethylmaleimide reductase | 39,492 | 5.8 | 7 | 52 | 19.7 |
| | | YaiI (excl.) | G6230 | DUF188 domain-containing protein YaiI | 16,958 | 5.4 | 5 | 18 | 22.4 |
| 816 | 3.0 ± 0.7 | PyrC (excl.) | EG10806 | Dihydroorotase | 38,802 | 5.7 | 17 | 131 | 21.8 |
| | | YqhD | G7564 | NADPH-dependent aldehyde reductase YqhD | 42,070 | 5.7 | 11 | 65 | 21.7 |

**TABLE 5**  List of marker protein spots induced after treatment with auranofin in *E. coli* MG1655[a] (*Continued*)

| Spot ID | Regulation factor ± range | Protein | EcoCyc accession number | Function | Theoretical MW [Da] | Theoretical pI | Peptides | Fragments | Coverage % |
|---------|---------|---------|---------|---------|---------|---------|---------|---------|---------|
| 852 | 3.0 ± 0.8 | YoaA (excl.) | G6992 | ATP-dependent DNA helicase YoaA | 70,333 | 6.7 | 5 | 21 | 6.4 |
| | | PstS (excl.) | EG10734 | Phosphate ABC transporter periplasmic binding protein | 37,001 | 9.0 | 3 | 12 | 9.0 |
| | | DnaE (excl.) | EG10238 | DNA polymerase III subunit α | 129,822 | 5.0 | 5 | 22 | 3.2 |
| | | PaaE (excl.) | G6713 | Phenylacetyl-CoA 1,2-epoxidase, reductase subunit | 39,294 | 5.9 | 3 | 11 | 4.8 |
| 859 | 3.6 ± 1.4 | **PykF** | EG10804 | Pyruvate kinase 1 | 50,697 | 5.7 | 10 | 61 | 17.2 |
| | | AhpF (excl.) | EG11385 | Alkyl hydroperoxide reductase, AhpF component | 56,142 | 5.3 | 3 | 17 | 5.8 |
| 860 | 3.6 ± 0.8 | **AhpF** | EG11385 | Alkyl hydroperoxide reductase, AhpF component | 56,142 | 5.3 | 22 | 168 | 36.5 |
| | | **LeuA** | EG11226 | 2-isopropylmalate synthase | 57,262 | 5.4 | 19 | 126 | 31.5 |
| | | PyrF (excl.) | EG10809 | Rotidine-5′-phosphate decarboxylase | 26,333 | 5.8 | 4 | 14 | 24.9 |

[a]Bold proteins met criteria for MS identification, proteins denoted with (excl.) were matched by the search engine, but were excluded, as they did not meet the criteria for Peptides (at least 3), Fragments (at least 10), or gel position was inconsistent with theoretical MW or pI. N/A: the search engine did not produce a match. EcoCyc accession number and function obtained from the EcoCyc database (43). Coverage % (at least 5), Coverage: coverage of the protein sequence by the annotated peptides. Fragments: number of product ions annotated by the search engine. Coverage: coverage of the protein sequence by the annotated peptides. Peptides: number of peptides annotated by the search engine. Fragments: number of product ions annotated by the search engine. Coverage: coverage of the protein sequence by the annotated peptides in %.

**TABLE 6** MICs of auranofin against *E. coli* BW25113 and *E. coli* BW25113 D*tolC*

| Strain | MIC, µg/mL |
|---|---|
| *E. coli* BW25113 (WT) | 128 |
| *E. coli* JW5503 (BW25113 Δ*tolC*) | 64 |

compounds and conferring resistance to the cell. The outer membrane protein TolC, part of the small multidrug resistance (SMR) protein family, is a significant component involved in the export of various cytotoxic substances. Studies have shown that mutations in the TolC protein resulted in sensitivity to different antibiotics, such as vancomycin, erythromycin, and novobiocin (48), and an increased susceptibility to metal ions (49).

We found that cellular susceptibility to auranofin was increased twofold when we tested the MIC of an *E. coli* Δ*tolC* strain (Table 6). The MICs of *B. subtilis* and *S. aureus* were 0.5 µg/mL, that is, still more than two orders of magnitude lower than that of the TolC-deficient *E. coli* strain. This suggests that TolC does play a role in the efflux of auranofin from the bacterial cell, albeit not to an extent that would fully explain the overall higher resistance of Gram-negative bacteria.

## Auranofin inhibits TrxB from both *E. coli* and *B. subtilis* to a similar extent

Evidence suggests that auranofin primarily exerts its effect by targeting bacterial thiol-redox homeostasis by inhibiting thioredoxin reductase (TrxB) (22, 24). Since TrxB is responsible for reducing oxidized thioredoxin (TrxA) and, thus, for maintaining a reducing intracellular environment, loss of TrxB activity can lead to severe oxidative stress in an organism with no alternative antioxidant system, such as the glutathione/glutaredoxin system.

While auranofin has been found to effectively inhibit TrxB of *S. aureus*, *M. tuberculosis,* and other Gram-positive bacterial species (22, 29), less is known about its inhibitory potency toward TrxB of Gram-negative bacteria. Hence, one possible explanation for auranofin's limited antibacterial activity against Gram-negative bacteria might be that auranofin is a less effective inhibitor of TrxB in these organisms when compared to Gram-positive bacteria. To explore this hypothesis, we tested auranofin's effectiveness in a biochemical assay with TrxB from Gram-negative and Gram-positive bacteria, with *E. coli* and *B. subtilis* as representative organisms, respectively.

The activity of purified TrxB from *E. coli* and *B. subtilis* (from now on referred to as "*ec*TrxB" and "*bs*TrxB," respectively) was then determined in the presence of NADPH, and various auranofin concentrations by performing a 5,5′-dithiobis(2-nitrobenzoic) acid (DTNB) reduction assay (50). The NADPH-dependent TrxB reduces DTNB in this assay, yielding the detectable 5-thio-2-nitrobenzoic acid (TNB). The amount of liberated TNB can be quantified spectrophotometrically by measuring its absorbance at 412 nm.

When auranofin was preincubated with 1 µM *ec*TrxB or *bs*TrxB, auranofin inhibited both enzymes dose dependently. The activity measurements showed potent and nearly complete inhibition of *ec*TrxB and *bs*TrxB already at equimolar (1 µM auranofin, corresponding to ~0.68 µg/mL) (Fig. 2). While the auranofin concentration required to fully inhibit TrxB agreed well with the previously determined MIC against *B. subtilis* (Table 1), it was far below the level required to exhibit cytotoxicity toward *E. coli*. This observation strongly suggests that the lower antimicrobial activity of auranofin against *E. coli* is not due to a lower affinity of auranofin for *ec*TrxB.

## Trx reductase is not the only target of auranofin in Gram-negative bacteria

Based on the results of our biochemical activity assays, we suspected that TrxB may not be the only target of auranofin in *E. coli*. We speculated that other redox pathway components might also be involved in the mechanism of action of auranofin. To test our hypothesis, we analyzed auranofin's antimicrobial activity against additional mutants of the KEIO collection (30). The mutants we chose lack components of the Trx antioxidant

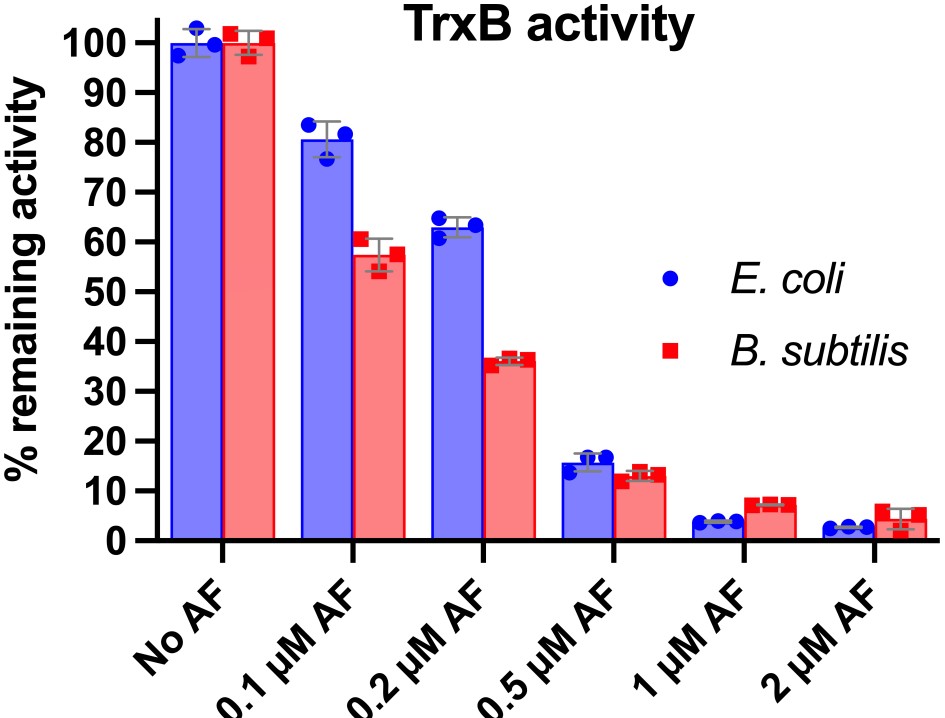

**FIG 2** Auranofin inhibits *E. coli* (*ec*TrxB) and *B. subtilis* TrxR (*bs*TrxB) in a dose-dependent manner. Purified *ec*TrxB and *bs*TrxB (1 µM) were preincubated with 200 µM NADPH in the absence (0 µM auranofin) or presence of various auranofin concentrations (0.1–2 µM) for 5 min. The addition of 2 mM DTNB initiated the Reaction. Reaction progress was monitored by measuring the absorbance of the liberated TNB chromophore at 412 nm over time. Experiments were performed in triplicates. Mean (bars), individual values (dots), and standard deviation are shown (error bars).

system, that is, the redox protein thioredoxin TrxA (Δ*trxA; E. coli* JW5156) or TrxB (Δ*trxB; E. coli* JW0871) itself, respectively (Fig. 3). Growth of the selected *E. coli* strains in the absence of auranofin was monitored until an OD$_{600}$ of 0.5 was reached. *E. coli* mutant strains without a functional Trx system showed a comparable or only slightly lower growth rate than the wild type, excluding the possibility that potential differences in the susceptibility of the different strains to auranofin are solely due to a generally impaired growth of the deletion mutants (Fig. 3a). The cultures were then used to inoculate medium containing different amounts of auranofin for a MIC assay. After overnight incubation, cultures were analyzed for growth. *E. coli* deficient in TrxA or TrxB showed a fourfold increased sensitivity to auranofin compared to wild-type *E. coli*, with MICs of 32 µg/mL (Δ*trxA* and Δ*trxB*) in contrast to 128 µg/mL (WT) (Fig. 3b and c). The increased susceptibility of *E. coli* mutants lacking a functional Trx system, especially the suspected target TrxB, indicates that auranofin leads to global oxidative stress not only through TrxB inhibition but also in a direct manner or through inhibition of other thiol-disulfide oxidoreductase systems.

**The glutathione reductase confers no resistance to Auranofin, while the glutathione pool by itself has a substantial effect on the MIC**

Although auranofin showed a generally higher efficacy against *E. coli* strains deficient in the functional Trx system, the MIC values were still markedly higher than those observed with Gram-positive bacteria. In contrast to most Gram-negative bacteria, many Gram-positive species lack glutathione. Glutathione by itself is considered an antioxidant, and with the small redox protein glutaredoxin, it can compensate for the loss of a functional Trx system (51). In Gram-positive organisms, on the other hand, the Trx system is essential (17, 52, 53). Hence, the additional glutathione/glutaredoxin system in

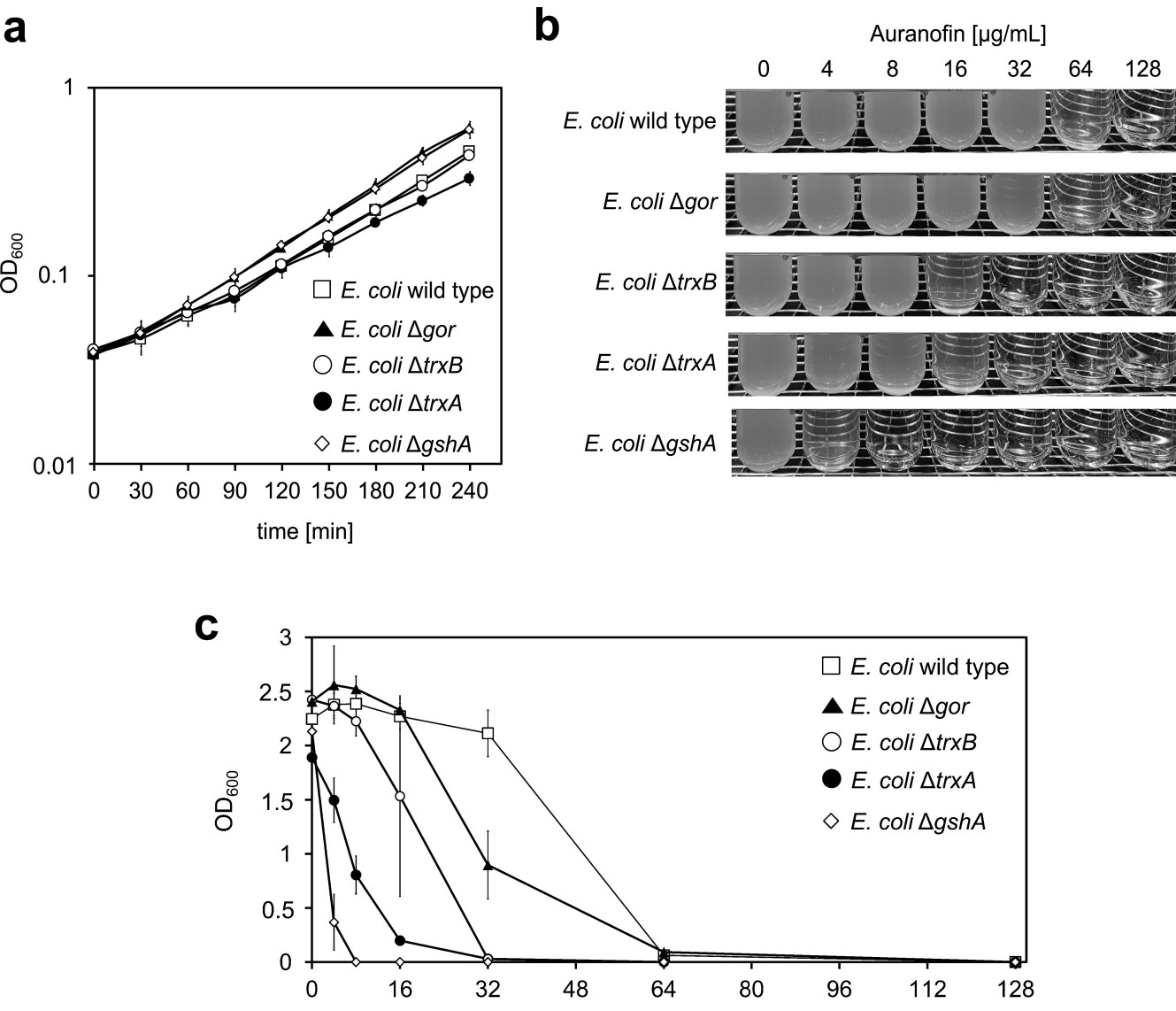

**FIG 3** Loss of a functional Trx system and depletion of the glutathione pool increase *E. coli*'s susceptibility to auranofin. (a) Growth curves of *E. coli* BW25113 (wild type) and single-gene knockout mutants of the KEIO collection (30) lacking a functional *gor* (Δ*gor*; *E. coli* JW3467), *gshA* (Δ*gshA*; *E. coli* JW2663), *trxB* (Δ*trxB*; *E. coli* JW0871), or *trxA* gene (Δ*trxA*; *E. coli* JW5156), respectively, in the absence of auranofin. (b and c) MICs of auranofin against *E. coli* wild-type and the various mutant strains. Cells were grown in an MOPS minimal medium until an $OD_{600}$ of 0.5 was reached. The various cultures were then used to inoculate MOPS minimal medium containing the indicated amounts of auranofin with $10^5$ cells. After overnight incubation, bacterial growth was analyzed (b) visually and (c) photometrically by measuring the $OD_{600}$ of the cultures. (b) A representative MIC assay is shown. (c) Means and standard deviations of three independent experiments are shown.

Gram-negative bacteria could be a reason for auranofin's low potency against *E. coli* and other Gram-negative species. However, we already knew from our MIC tests performed in mutants deficient in proteins upregulated in our proteomics experiments that a Δ*gor* mutation did not affect the MIC. Thus, we argued that the absence of a functional glutaredoxin system does not play a role. We then tested whether glutathione, the tripeptide itself, could play an important role instead. Both, the deletion of glutathione oxidoreductase (Δ*gor*) and γ-glutamate-cysteine-ligase (Δ*gshA*), the enzyme catalyzing the first step of glutathione biosynthesis, did not affect *E. coli*'s growth in the absence of auranofin (Fig. 3a). But auranofin's activity against *E. coliΔgshA* was markedly higher

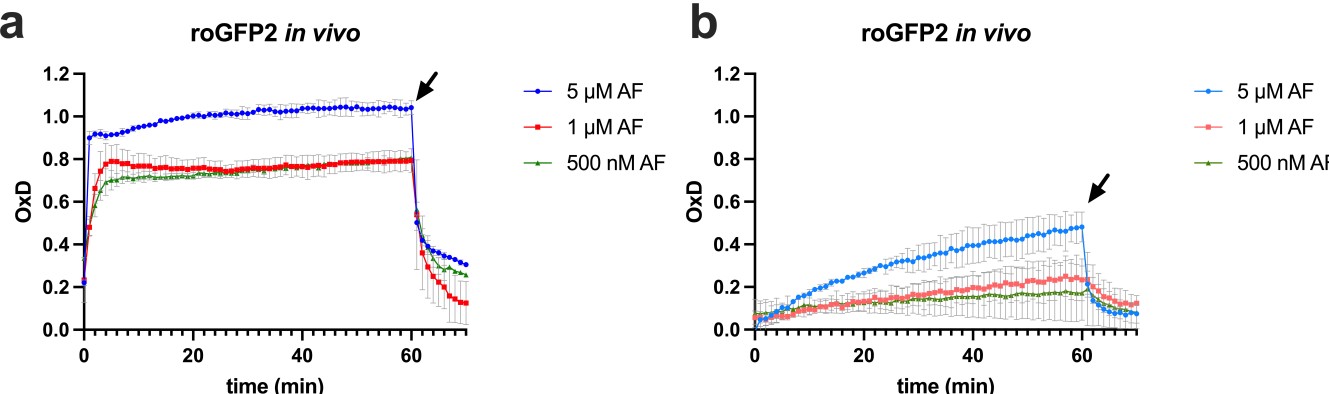

**FIG 4** Auranofin exposure leads to roGFP2 oxidation *in vivo*. To evaluate the effects of auranofin on protein thiols in bacteria *in situ*, *B. subtilis* (a) and *E. coli* (b) expressing roGFP2 were treated with 0.5, 1, or 5 µM auranofin. Aldrithiol-2 and dithiothreitol were used to fully oxidize and fully reduce the probe within cells, respectively, and these baselines (not shown) were used to calculate the oxidation degree (OxD) of roGFP2. OxD normalizes the ratiometrically determined redox state and accounts for differences in probe synthesis and measurement settings. roGFP2 fluorescence was measured at an emission wavelength of 510 nm, and the excitation wavelength was scanned from 350–500 nm. roGFP2 oxidation was monitored for 60 min after adding auranofin, and then DTT was added at a concentration of 2 mM (indicated by an arrow). Means and standard deviations are shown.

than against the WT or Δ*gor*, which both had the same MIC. The MIC of Δ*gshA* decreased to 8 µg/mL (Fig. 3b and c), closer to the MIC observed in Gram-positive bacteria. The increase in *E. coli's* sensitivity to auranofin upon depletion of the intracellular glutathione pool suggests that this monothiol, typically present at concentrations in the millimolar range, confers substantial protection against oxidative stress induced by auranofin.

## Auranofin leads to thiol oxidation in the cytosol

We then wanted to measure the intracellular thiol redox state in auranofin-treated bacteria *in situ*. roGFP2 is a genetically engineered variant of GFP containing two cysteines, which form a disulfide bridge upon oxidation, thereby changing the protein's fluorescent properties. This can be used to determine the oxidation degree of roGFP2 (54), independent of the probe's concentration (33, 55, 56). Because we assumed that auranofin interferes not only with TrxB but with the oxidation state of thiols in general, we used roGFP2 as a tool to monitor the redox state of cysteines in cellular proteins. We expressed roGFP2 in the cytoplasm of both *B. subtilis* and *E. coli*. We then determined the oxidation degree of roGFP2 inside the cells. The oxidant AT-2 was used to fully oxidize, and the reductant DTT to fully reduce roGFP2 within cells, setting the fully reduced and fully oxidized baselines of our probe within the bacterial cell. In both strains, the probe is more or less fully reduced in the cytosol of untreated cells. Using a concentration of 5 µM auranofin, the degree of oxidation of roGFP2 increases by 40% after 60 min of treatment in *E. coli* (Fig. 4b), whereas it increases by up to 90% in less than 1 min in *B. subtilis* (Fig. 4a). Lower concentrations of 1 µM or 500 nM auranofin still increased the oxidation state of roGFP2 in *B. subtilis* to around 80% within 5 min of treatment, while in *E. coli* an oxidation state of only 10%–20% was reached after 60 min. The subsequent addition of 2 mM DTT to the cells led to an immediate decrease in probe oxidation in all cases, demonstrating that, in principle, the change in cellular oxidation imposed by auranofin is reversible by thiol reductants in both organisms.

## DISCUSSION

The gold(I) compound auranofin, an FDA-approved antirheumatic drug, has recently been found to possess antimicrobial activity and, thus, is a promising candidate for drug repurposing for treating multidrug-resistant infections. Although auranofin shows high activity against a considerable range of Gram-positive bacteria, it seems less effective

against Gram-negative species (22, 29). In the present study, we aimed to understand the underlying mechanism of the difference in susceptibility of Gram-negative and Gram-positive bacteria to auranofin.

We thus tested auranofin's ability to inhibit the growth of different Gram-positive and Gram-negative species. Growth of Gram-positive strains was prevented by an auranofin concentration as low as 0.5 µg/mL. These results agree with previously reported MICs against *B. subtilis* PY79*, M. tuberculosis,* and several MRSA strains (22, 24). As expected, a substantial difference in the effective concentration of auranofin was found for the Gram-negative species *A. baumannii* DSM 30007, *E. coli* DSM 30083, and *P. aeruginosa* DSM 50071 with MIC values of 32, 128, and 512 µg/mL, respectively.

Proteomic profiling of auranofin-stressed *E. coli* revealed that its cellular response is similar to the response of *B. subtilis*. *B. subtilis* was found to counteract auranofin stress by specifically upregulating enzymes that detoxify ROS, such as the catalase KatA, or replenish the cysteine pool, such as YrhB (42). Similarly, in *E. coli*, proteins belonging to the OxyR regulon, such as AhpC and AhpF, the two components of *E. coli*'s alkyl hydroperoxide reductase and glutathione oxidoreductase Gor, were upregulated significantly. The transcription factor OxyR regulates the expression of antioxidant genes in response to oxidative stress, particularly elevated levels of hydrogen peroxide (44, 45). We showed previously that OxyR-dependent genes are also upregulated in response to the oxidant diamide and the antimicrobial compound allicin, both of which were found to induce disulfide stress in *E. coli* (46, 47). We then tested the susceptibility of mutants devoid of proteins upregulated in our proteomic experiments to auranofin. None of the mutants tested showed a lower MIC and, thus, higher susceptibility to auranofin, suggesting that these proteins are not the reason for *E. coli*'s high tolerance for auranofin.

Previous studies on auranofin as an antimicrobial drug suggested that, as in mammals, the flavoenzyme thioredoxin reductase (TrxB) is a primary target in bacteria (22). Our studies showed that auranofin inhibits mammalian TrxR with a $K_i$ of only 4 nM (57). Along this line, auranofin has been shown to effectively inhibit bacterial TrxB from numerous Gram-positive pathogens, including *M. tuberculosis* and *S. aureus* (22, 58). However, it should be noted that bacterial TrxB is structurally significantly different from mammalian TrxR and does not contain selenocysteine (17). Selenocysteine is thought to play a major role in the inhibition of mammalian TrxR by auranofin: Several lines of evidence suggest that selenocysteine is either the direct binding site of auranofin or is essential for releasing the gold atom from its ligands, given that the affinity of selenium for gold and its nucleophilic power are greater than those of sulfur (13, 59–62). Indeed, there is direct evidence for auranofin binding to the selenocysteine in human TrxR (63). In addition, TrxBs from Gram-positive and Gram-negative bacteria are structurally much more similar to each other than to mammalian TrxR. Bacterial TrxB has only a low degree of sequence homology (<30%) to human TrxR (61).

Nevertheless, to uncover potential differences in TrxB inhibition in Gram-negative and Gram-positive bacteria, which could explain the different susceptibilities of the species, we biochemically tested the inhibition of TrxB from *E. coli* (*ec*TrxB) and *B. subtilis* (*bs*TrxB) by auranofin. We found that inhibition of both *ec*TrxB and *bs*TrxB by auranofin is dose dependent, and in both cases, an equimolar amount is enough to inhibit their activity entirely. These observations suggest that inhibition of bacterial TrxB by auranofin is irreversible. In any case, no significant difference in susceptibility could be observed between TrxB from Gram-positive and Gram-negative species in our assay. This shows that, at a target level, both Gram-positive and Gram-negative species are equally susceptible.

To test other properties that could make Gram-negative bacteria more resistant to auranofin, we examined an *E. coli* strain lacking TolC. This efflux pump removes antibiotics and metals from the cell (64). We found that the lack of TolC decreased auranofin's resistance only by a factor of two, not fully explaining Gram-negative organisms' substantially higher resistance than Gram-positive bacteria.

Unlike most Gram-negative bacteria, many Gram-positive bacteria do not possess the low-molecular-weight thiol glutathione and the glutaredoxin system. Hence, TrxB and its substrate thioredoxin (TrxA) are considered the key mediators of redox homeostasis in these organisms (17), and the Trx system is essential for *B. subtilis* and *S. aureus*' survival at normal growth conditions and under oxidative stress (52, 53). Conversely, *E. coli* strains deficient in TrxB or TrxA did show only moderately higher susceptibility toward auranofin. These findings, too, suggest that TrxB may not be the sole target of auranofin in *E. coli*. Following up on the idea that glutathione and the associated glutaredoxins could compensate for the loss of TrxB in *E. coli*, as suggested by Harbut et al. (22), we revisited auranofin's activity against an *E. coli* mutant devoid of the glutathione reductase (Gor). As there was no difference in the susceptibility to auranofin, in line with findings from Thangamani et al. (27), we concluded that the glutaredoxin system as a backup for thioredoxin is not the main reason for *E. coli*'s relative resistance to auranofin. However, the loss of the major intracellular low-molecular-weight thiol glutathione itself led to a substantial increase in sensitivity to auranofin. Similar observations were recently made with auranofin analogs in *Burkholderia cenocepacia* (65). Therefore, several possible factors for the high tolerance of Gram-negative bacteria toward auranofin must exist. First and foremost, Gram-negative bacteria possess more effective mechanisms to avoid auranofin stress, that is, the presence of efflux pumps, in combination with the presence of an outer membrane. Second, the presence of glutathione was, in our experiments, the most effective contributor to auranofin resistance in *E. coli*. Glutathione's high concentration of up to 10 mM in the cell could make it a potent buffer against auranofin, depleting the gold compounds' ability to react with protein thiols. These findings corroborate that auranofin induces disulfide stress not only through TrxB inactivation but potentially through direct thiol modification in proteins and reactions with low-molecular-weight thiols in general.

The cysteine-containing redox probe roGFP2, synthesized in the cytosol of bacterial cells, revealed that auranofin indeed directly affected protein oxidation. A high oxidation degree was reached within minutes in Gram-positive bacteria. This quick oxidation is not observed in Gram-negative bacteria, presumably due to the existence of efflux pumps in combination with the presence of high concentrations of glutathione.

In clinical trials with auranofin, plasma concentrations of auranofin ranging between 0.8 and 1.5 µg/mL were reached (66). While this therapeutically achievable auranofin concentration is sufficient to inhibit the growth of most Gram-positive pathogens, it is 20–340 times lower than the auranofin concentration necessary to prevent the growth of the tested Gram-negative strains, rendering auranofin virtually ineffective against these organisms in a clinical setting. Its limited activity against Gram-negative bacteria currently prevents its systemic clinical use as an antibiotic in treating infections with Gram-negative pathogens. The results presented here strongly suggest that, while auranofin's mechanism of action against *E. coli* does not substantially differ from that against *B. subtilis per se*, other factors unique to Gram-negative bacteria are attenuating auranofin's antimicrobial effectiveness. Thus, potential drugs based on auranofin's mechanism of action need to account for those factors, particularly the presence of high cellular concentrations of glutathione.

Our results indicate that the mode of action of auranofin in bacteria is a combination of the inactivation of thiol-containing enzymes, a decrease in reduced Trx levels through TrxB inactivation, and the subsequent induction of systemic oxidative stress. While most clinically used antibiotics act on a single target, this broad-range effect of auranofin could prevent the evolution of specific resistance mechanisms, and indeed several others reported that spontaneous auranofin-resistant mutants cannot be generated (22, 27, 67). This should make the development of other gold-containing drugs, effective in Gram-negative bacteria, and based on auranofin's mode of action a worthwhile endeavor.

## ACKNOWLEDGMENTS

L.I.L. and J.E.B. acknowledge funding from the German Research Foundation (DFG) through Research Training Group 2341 "Microbial Substrate Conversion (MiCon)."

L.I.L. received additional funding through the InnovationsFoRUM Host-Microbe-Interaction IF-018–22-TP8, and J.E.B. gratefully acknowledges funding from the German federal state of North Rhine-Westphalia for the mass spectrometer (Forschungsgrossgeräte der Länder).

We gratefully acknowledge the excellent support of the technical staff at RUBION, Ruhr University Bochum.

J.E.B. and L.I.L. conceived and coordinated the study. A.U., A.M.I., M.V., and N.L. purified all proteins. M.F. and K.B. provided materials and developed protocols for the biochemical activity and inhibition assays. M.V-H. performed the experiments shown in Fig. 1. L.Q.B. performed the experiments shown in Fig. 2. A.M.I., M.V., and L.Q.B. performed the experiments shown in Fig. 3. L.Q.B. and J.Y. performed the experiments shown in Fig. 4. A.U., L.Q.B., M.V., M.V.-H., J.E.B., and L.I.L. contributed to the analysis and interpretation of the results. A.U., L.Q.B., and L.I.L. wrote the manuscript. All authors approved the final version of the manuscript.

## AUTHOR AFFILIATIONS

[1]Medical Faculty, Institute of Biochemistry and Pathobiochemistry–Microbial Biochemistry, Ruhr University Bochum, Bochum, Germany

[2]Faculty of Biology and Biotechnology, Applied Microbiology, Ruhr University Bochum, Bochum, Germany

[3]Institute of Electrical Engineering and Applied Sciences–Molecular Biology, Westphalian University of Applied Sciences, Recklinghausen, Germany

[4]Interdisciplinary Research Center, Justus Liebig University Giessen, Giessen, Germany

## PRESENT ADDRESS

Agnes Ulfig, University of Iceland, Faculty of Medicine, Louma G. Laboratory of Epigenetic Research, Reykjavik, Iceland

## AUTHOR ORCIDs

Lars I. Leichert http://orcid.org/0000-0002-5666-9681

## FUNDING

| Funder | Grant(s) | Author(s) |
| --- | --- | --- |
| Deutsche Forschungsgemeinschaft (DFG) | RTG2341 | Julia E. Bandow |
| | | Lars I. Leichert |
| RUB \| Medizinische Fakultät, Ruhr-Universität Bochum (Faculty of Medicine at Ruhr-Universität Bochum) | IF-018-22 | Lars I. Leichert |
| Land Nordrhein-Westfalen (NRW) | Forschungsgrossgeräte der Länder | Julia E. Bandow |

## ADDITIONAL FILES

The following material is available online.

Open Peer Review

**PEER REVIEW HISTORY (review-history.pdf).** An accounting of the reviewer comments and feedback.

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
