## [Reviewer comments · Microbiology Spectrum]

Microbiology Spectrum

Comparison of the mechanism of antimicrobial action of the gold(I) compound auranofin in Gram-positive and Gram-negative bacteria

Laisa Quadros Barse, Agnes Ulfig, Marharyta Varatnitskaya, Melissa Vázquez Hernández, Jihyun Yoo, Astrid Imann, Natalie Lupilov, Marina Fischer, Katja Becker, Julia Bandow, and Lars Leichert

Corresponding Author(s): Lars Leichert, Ruhr-Universität Bochum

Review Timeline:

Submission Date:	January 15, 2024
Editorial Decision:	May 3, 2024
Revision Received:	July 1, 2024
Accepted:	August 13, 2024

Editor: Brian Conlon

Reviewer(s): Disclosure of reviewer identity is with reference to reviewer comments included in decision letter(s). The following individuals involved in review of your submission have agreed to reveal their identity: Fahim Hayat (Reviewer #2)

Transaction Report:

DOI: <https://doi.org/10.1128/spectrum.00138-24>

Re: Spectrum00138-24 (Comparison of the mechanism of antimicrobial action of the gold(I) compound auranofin in Gram-positive and Gram-negative bacteria)

Dear Dr. Lars I Leichert:

Thank you for the privilege of reviewing your work. Below you will find my comments, instructions from the Spectrum editorial office, and the reviewer comments. There are numerous important comments raised by the reviewers that need to be addressed before this manuscript can be considered for publication.

Revision Guidelines

Sincerely,
Brian Conlon
Editor
Microbiology Spectrum

Reviewer #1 (Comments for the Author):

The authors submit a manuscript aiming to determine the mechanism driving the differential susceptibility of auranofin towards Gram-positive and Gram-negative pathogens. There is a continued interest in using repurposed FDA-approved agents as small molecules to counter rising antimicrobial resistance. Auranofin is FDA-approved for the treatment of rheumatoid arthritis and has

been previously shown to possess antimicrobial activity against Gram-positive and Gram-negative pathogens. The MIC towards Gram-negative pathogens is >100x higher compared to Gram-positive pathogens, which is the focus of this study. There have been several studies examining the antimicrobial mechanism of auranofin against Gram-positive and Gram-negative pathogens previously, with the target in *E. coli* thought to be TrxB. However, it is thought to have other targets given the inability of others to generate auranofin-resistant mutants.

The key findings of this manuscript are as follows:

1. Using proteomics, they show treatment of *E. coli* with subinhibitory concentrations of auranofin leads to upregulation of several genes associated with oxidative stress response
2. The MIC versus a tolC mutant of *E. coli* reduces the MIC by just two fold, indicating efflux unlikely the sole mechanism of auranofin resistance in *E. coli*
3. Auranofin inhibits purified TrxB (the auranofin target proposed by previous studies) equally, indicating differences in target binding does not explain the differential susceptibility.
4. Knockouts in *gshA*, *trxA* and *trxB* (but not *gor*) decrease the MIC of auranofin by 2-6x (corroborates published data by Harbut et al PNAS 2015), which is proposed by this group and others to be due to depletion of intracellular glutathione pool
5. Similar concentrations of auranofin cause significantly less thiol oxidation in *E. coli* versus *B. subtilis*, which the authors attribute to smaller glutathione pool in Gram-positives when compared to Gram-negatives.

General comments:

There are several existing studies examining the mechanism of auranofin as an antimicrobial agent in Gram-negative and Gram-positive bacteria (Thangamani, Sci Rep, 2016, Maydaniuk, mSpectrum, 2024, Harbut, PNAS, 2015). It is generally accepted that *trxB* is the primary target auranofin and the redundant glutathione/thioredoxin system is the basis for the decreased susceptibility in *E. coli* (Harbut, PNAS, 2015). While this manuscript corroborates the findings of these other groups (including *gshA* deletion decreasing auranofin MIC in *E. coli*), my main concern is that it does not substantially add to the existing body of literature.

Major comments:

1. One common barrier to drugs in Gram-negative bacteria is the outer membrane, which is absent in Gram-positive bacteria. Other investigators have attributed this key structural difference to the decreased susceptibility in *E. coli* (Thangamani, Sci Rep, 2016). This would explain the higher MIC and the lower level of thiol oxidation in *E. coli* vs *B. subtilis*. This should be addressed in this study to provide meaningful mechanistic insight into Gram-negative resistance to Auranofin. The authors should perform an experiment to show that the permeability of Gram-negative bacteria to auranofin is comparable to Gram-positive bacteria to support their proposed mechanism and exclude the outer membrane barrier as the main driver of Gram-negative resistance.
2. If the authors are proposing the MOA of auranofin is ROS-mediated cell death, they should perform an experiment using a ROS-sensitive probe (e.g H₂DCFDA) to show auranofin treatment increases ROS production.
3. The authors show depletion of the intracellular glutathione pool by deletion of *gshA* decreases the MIC to auranofin. It may be helpful to show that addition of exogenous glutathione to the WT and *gshA* knockout increases the MIC.

Minor comments:

1. Please calculate the IC₅₀ to quantify the inhibitory properties of aureolysin against *E. coli* and *S.aureus trxB*
2. Fib 3b/c - why were the MICs performed in MOPs rather than by CLSI standards as in table 4?
3. Line 452: should read "resistance" instead of "tolerance", given that the MIC is increased versus antibiotic tolerance, which refers to decreased bactericidal activity
4. Table 1: Several strains referred to as "type strain", should this be "wild type strain"?

Reviewer #2 (Comments for the Author):

The manuscript presents a well-structured study on the differential effects of the gold(I) compound auranofin on Gram-positive and Gram-negative bacteria. It leverages robust methodologies including MIC determination, proteomic analysis, and enzymatic assays, providing a sound basis for reproducible and rigorous experimental science. However, to ensure the submission aligns perfectly with the high standards of Microbiology Spectrum, focusing on technical and methodological rigor, several enhancements are recommended. Firstly, while the methods are generally well-described, the manuscript could benefit from additional details concerning the experimental conditions, such as temperature specifics, bacterial strain handling, and the exact concentrations of auranofin used across different assays. These details are crucial for reproducibility, especially in multi-lab settings. It would also be advantageous to expand on the controls used in each experiment, particularly detailing any vehicle controls for auranofin and describing the statistical rationale for the number of biological and technical replicates chosen for each experimental condition.

Moreover, the statistical analysis section requires further elaboration. The authors should specify the statistical tests applied to each dataset, the justification for their selection, and discuss the data distribution assumptions checked before applying these tests. Including a more detailed analysis of the data, such as interaction effects or dose-response relationships, could significantly enhance the depth of the study. Presentation of data also needs attention; ensuring that all figures and tables clearly display variability measures such as standard deviations or confidence intervals would aid in the transparency and understanding of the results.

Additionally, considering the complex nature of proteomic analysis, it is recommended that the authors validate some of the key findings using an independent method such as Western blotting for selected proteins to confirm changes in protein expression levels observed in the proteomic studies. This would not only strengthen the validity of the results but also provide a clearer link between observed proteomic changes and phenotypic outcomes. Furthermore, addressing potential limitations in the methods used and discussing their implications on the results would provide a more balanced view and suggest directions for future research.

Lastly, ensuring compliance with all relevant biosafety and ethical guidelines is crucial, especially when manipulating bacterial strains. A statement regarding such compliance should be included if not already present. These enhancements will solidify the manuscript's contribution to the field, ensuring technical excellence and methodological rigor as required by Microbiology Spectrum.

POINT-BY-POINT REPLY TO THE REVIEWERS' CONCERNS

We thank the reviewers for their insightful comments. We have now re-written the manuscript to address their concerns point-by-point, as outlined below. Our responses to the reviewers' criticisms are colored in green. In our responses, we point to the line numbers in which these responses were incorporated into the revised manuscript. Additionally, we have highlighted in yellow the revised sections in a "Marked Up Manuscript - For Review Only File".

The authors submit a manuscript aiming to determine the mechanism driving the differential susceptibility of auranofin towards Gram-positive and Gram-negative pathogens. There is a continued interest in using repurposed FDA-approved agents as small molecules to counter rising antimicrobial resistance. Auranofin is FDA-approved for the treatment of rheumatoid arthritis and has been previously shown to possess antimicrobial activity against Gram-positive and Gram-negative pathogens. The MIC towards Gram-negative pathogens is >100x higher compared to Gram-positive pathogens, which is the focus of this study. There have been several studies examining the antimicrobial mechanism of auranofin against Gram-positive and Gram-negative pathogens previously, with the target in *E. coli* thought to be TrxB. However, it is thought to have other targets given the inability of others to generate auranofin-resistant mutants.

The key findings of this manuscript are as follows:

1. Using proteomics, they show treatment of *E. coli* with subinhibitory concentrations of auranofin leads to upregulation of several genes associated with oxidative stress response
2. The MIC versus a tolC mutant of *E. coli* reduces the MIC by just two fold, indicating efflux unlikely the sole mechanism of auranofin resistance in *E. coli*
3. Auranofin inhibits purified TrxB (the auranofin target proposed by previous studies) equally, indicating differences in target binding does not explain the differential susceptibility.
4. Knockouts in *gshA*, *trxA* and *trxB* (but not *gor*) decrease the MIC of auranofin by 2-6x (corroborates published data by Harbut et al PNAS 2015), which is proposed by this group and others to be due to depletion of intracellular glutathione pool
5. Similar concentrations of auranofin cause significantly less thiol oxidation in *E. coli* versus *B. subtilis*, which the authors attribute to smaller glutathione pool in Gram-positives when compared to Gram-negatives.

General comments:

There are several existing studies examining the mechanism of auranofin as an antimicrobial agent in Gram-negative and Gram-positive bacteria (Thangamani, Sci Rep, 2016, Maydaniuk, mSpectrum, 2024, Harbut, PNAS, 2015). It is generally accepted that *trxB* is the primary target auranofin and the redundant glutathione/thioredoxin system is the basis for the decreased susceptibility in *E. coli* (Harbut, PNAS, 2015). While this manuscript corroborates the findings of these other groups (including *gshA* deletion decreasing auranofin MIC in *E. coli*), my main concern is that it does not substantially add to the existing body of literature.

We are aware that perceived "novelty" is not the major strength of our manuscript. However, we decided to submit our manuscript to Microbiology Spectrum in part because it "welcomes all types of microbiology research studies [...] without consideration of novelty or impact".

We have cited and discussed the Thangamani et al. 2016 and the Harbut et al. 2015 studies in our original manuscript. We now also discuss the Maydaniuk et al. 2024 study, which used an auranofin analog against an auranofin-insensitive gram-negative species and was published in the February 6 issue of Microbiology Spectrum, after the submission of our manuscript (lines 659 f and 906 ff).

Major comments:

1. One common barrier to drugs in Gram-negative bacteria is the outer membrane, which is absent in Gram-positive bacteria. Other investigators have attributed this key structural difference to the decreased susceptibility in *E. coli* (Thangamani, Sci Rep, 2016). This would explain the higher MIC and the lower level of thiol oxidation in *E. coli* vs *B. subtilis*. This should be addressed in this study to provide meaningful mechanistic insight into Gram-negative resistance to Auranofin. The authors should perform an experiment to show that the permeability of Gram-negative bacteria to auranofin is comparable to Gram-positive bacteria to support their proposed mechanism and exclude the outer membrane barrier as the main driver of Gram-negative resistance.

In our original manuscript, we identified “several possible factors for the high tolerance of Gram-negative bacteria towards auranofin”. We specifically mention as the first one “the presence of efflux pumps, potentially in combination with the presence of an outer membrane.” We do not argue that the permeability of Gram-negative bacteria to auranofin is comparable to Gram-positive bacteria, quite the opposite. To clarify, we have now changed this part to read “First and foremost, Gram-negative bacteria possess more effective mechanisms to avoid auranofin stress, i.e., the presence of efflux pumps, in combination with the presence of an outer membrane.” (lines 661 ff).

We do agree that an experiment examining the permeability of the Gram-negative cell envelope components, such as the outer membrane, for auranofin would provide further insights. However, this would be a quite complex and time-consuming experiment, requiring technologies that are not yet established in our lab. We therefore feel that it is beyond the scope of the current study.

2. If the authors are proposing the MOA of auranofin is ROS-mediated cell death, they should perform an experiment using a ROS-sensitive probe (e.g H2DCFDA) to show auranofin treatment increases ROS production.

In our figure 4 we use the genetically encoded redox probe roGFP2 in both *B. subtilis* and *E. coli* to show auranofin-induced protein disulfide bond formation in the cytosol of bacteria. We do not argue that “ROS-mediated cell death” is the Mode of Action of auranofin. However, we stated “Our results indicate that the mode of action of auranofin in bacteria is a combination of the inactivation of thiol-containing enzymes, a decrease of reduced Trx levels through TrxB inactivation, and the induction of systemic oxidative stress.” We can see now how this might be perceived to mean that we consider “ROS-mediated cell death” to be the main factor of auranofin’s effectiveness, but, as we state later, we think auranofin has a broad-range effect on the cell on multiple levels. To clarify the fact that we do not propose that auranofin directly induces ROS-formation but rather think that oxidative stress within the cell is caused by the inactivation of thiol-containing enzymes and the decrease in thioredoxin activity, we have thus changed our concluding paragraph to read “the mode of action of auranofin in bacteria is a combination of the inactivation of thiol-containing enzymes, a decrease of reduced Trx levels through TrxB inactivation, and the subsequent induction of systemic oxidative stress.” (line 689).

3. The authors show depletion of the intracellular glutathione pool by deletion of *gshA* decreases the MIC to auranofin. It may be helpful to show that addition of exogenous glutathione to the WT and *gshA* knockout increases the MIC.

Since free thiols, such as glutathione can directly interact with auranofin, the external addition to the medium would probably inactivate auranofin even before it reaches the cell, thus we did not perform this experiment.

Minor comments:

1. Please calculate the IC₅₀ to quantify the inhibitory properties of aureolysin against *E. coli* and *S.aureus* *trxB*

We presume the reviewer meant auranofin instead of aureolysin and *B. subtilis* instead of *S. aureus*. A naïve approach of calculating the IC₅₀ from our data would yield an IC₅₀ in the range of 0.2-0.5 μM, roughly half the enzyme concentration used in our assay, for the enzymes of both species. However, if we presume that the inhibition of TrxB by auranofin is irreversible, this IC₅₀ value would depend on the concentration of enzyme present and thus any IC₅₀ would be meaningless (see e.g. Box 1 in Singh et al. 2011, The resurgence of covalent drugs. Nat Rev Drug Discov <https://doi.org/10.1038/nrd3410>). And our data, with full inhibition at equimolar concentrations, very much suggests that inhibition of TrxB is irreversible. To clarify our point, we now explicitly state in the discussion that “These observations suggest that inhibition of bacterial TrxB by auranofin is irreversible.” (line 636).

2. Fib 3b/c - why were the MICs performed in MOPs rather than by CLSI standards as in table 4?

We chose MOPS medium for two reasons. First, the proteomic studies were performed in this medium, as complex medium containing nutrients derived from protein sources, such as Mueller-Hinton CLSI Media, inhibit effective proteomic labeling with L-[³⁵S]-methionine. Second, as outlined in our response to the reviewer’s major comment #3, it is highly likely that glutathione and other free thiols directly interact with auranofin. Mueller-Hinton CLSI Media is

composed of complex nutrient sources that contain glutathione and presumably other free thiols, at concentrations that could vary from batch to batch. To clarify our rationale, we now added the sentence "To exclude the interference of antioxidative media components with auranofin, we performed these tests in the chemically defined MOPS medium, the same medium used for the proteomic L-[³⁵S]-methionine-labeling." (line 446 ff).

3. Line 452: should read "resistance" instead of "tolerance", given that the MIC is increased versus antibiotic tolerance, which refers to decreased bactericidal activity

We have changed the offending line (line 444).

4. Table 1: Several strains referred to as "type strain", should this be "wild type strain"?

These strains are indeed type strains, i.e. the reference strains designated as the nomenclatural "type" of that species. Which, as the reviewer notes, implies that they are wild type strains, as well.

Reviewer #2 (Comments for the Author):

The manuscript presents a well-structured study on the differential effects of the gold(I) compound auranofin on Gram-positive and Gram-negative bacteria. It leverages robust methodologies including MIC determination, proteomic analysis, and enzymatic assays, providing a sound basis for reproducible and rigorous experimental science. However, to ensure the submission aligns perfectly with the high standards of Microbiology Spectrum, focusing on technical and methodological rigor, several enhancements are recommended. Firstly, while the methods are generally well-described, the manuscript could benefit from additional details concerning the experimental conditions, such as temperature specifics, bacterial strain handling, and the exact concentrations of auranofin used across different assays. These details are crucial for reproducibility, especially in multi-lab settings. It would also be advantageous to expand on the controls used in each experiment, particularly detailing any vehicle controls for auranofin and describing the statistical rationale for the number of biological and technical replicates chosen for each experimental condition.

We have now clarified the Materials and Methods section.

- We explicitly added growth temperature and cell density for the MIC tests in MOPS minimal medium (lines 162 ff).
- We added a comment on routine strain handling (line 180 f).
- We explicitly mention all concentrations used in the MIC assays and TrxB inhibition assays individually (lines 169 f, 171, 255)
- We now explicitly point out our vehicle controls (line 172 f, line 355 f)
- We did not perform a full power analyses to justify the number of biological and technical replicates, but we feel that in most of our experiments, the extraordinarily large difference between the means (e.g. a two-fold increase in relative protein synthesis rate in our proteomic experiments (Fig. 1), a doubling of the MIC (Tables 4 and Figure 3), almost full inhibition of TrxB activity by the chosen concentrations of auranofin (Figure 2), an up to 5-fold rise in probe oxidation in auranofin-treated bacteria (Figure 4) justifies the sample sizes chosen.

Moreover, the statistical analysis section requires further elaboration. The authors should specify the statistical tests applied to each dataset, the justification for their selection, and discuss the data distribution assumptions checked before applying these tests. Including a more detailed analysis of the data, such as interaction effects or dose-response relationships, could significantly enhance the depth of the study. Presentation of data also needs attention; ensuring that all figures and tables clearly display variability measures such as standard deviations or confidence intervals would aid in the transparency and understanding of the results.

All of our figures already displayed standard deviations and we stated so in the figure legends, except for figure 2, where we abbreviated standard deviation as SD and figure 4, where we forgot to mention it. We have changed that now to clarify (line 943, line 964). We have now also added the range of the Regulation factor to Table 5.

Additionally, considering the complex nature of proteomic analysis, it is recommended that the authors validate some of the key findings using an independent method such as Western blotting for selected proteins to confirm changes in protein expression levels observed in the proteomic studies. This would not only strengthen the validity of the results but also provide a clearer link between observed proteomic changes and phenotypic outcomes.

If our proteomic experiments had measured protein abundance, such a validation could potentially provide valuable insights (although those experiments would be time-consuming and resource-intensive, requiring access to a large number of antibodies, many of which are not commercially available and would need to be generated first). However, in our proteomic experiments we detected newly synthesized proteins over a span of 5 minutes by radioactive labeling (i.e. protein synthesis rates), and such a detection is typically much more sensitive than a standard western blot. Especially for already highly abundant metabolic proteins, changes in protein synthesis need to accumulate over time to be picked up by western blots (which essentially determine protein abundance), due to the inherent limited dynamic range of the methodology. This is also not the first time that we have performed such an experiment, in fact we have determined changes in protein synthesis rates in response to antibiotics with the help of proteomics for well over 90 different compounds and while complex, our results have proven quite reliable (see e.g. Senges et al. 2020, *Antimicrobial Agents and Chemotherapy* doi:10.1128/AAC.01373-20)

Furthermore, addressing potential limitations in the methods used and discussing their implications on the results would provide a more balanced view and suggest directions for future research. Lastly, ensuring compliance with all relevant biosafety and ethical guidelines is crucial, especially when manipulating bacterial strains. A statement regarding such compliance should be included if not already present. These enhancements will solidify the manuscript's contribution to the field, ensuring technical excellence and methodological rigor as required by *Microbiology Spectrum*.

Based on the ASM guidelines we were under the impression that such statements must be made in the cover letter, and only when there are possible biosafety and biosecurity concerns (cf. <https://journals.asm.org/dual-use-research>). Since we used BSL1 laboratory strains of both *E. coli* and *B. subtilis*, we feel that there are no biosafety, biosecurity or ethical concerns whatsoever. However, we have now included the statement "All experiments were performed in compliance with German laws regarding biosafety and biosecurity." (line 181 f).

Re: Spectrum00138-24R1 (Comparison of the mechanism of antimicrobial action of the gold(I) compound auranofin in Gram-positive and Gram-negative bacteria)

Dear Dr. Lars I Leichert:

Your manuscript has been accepted, and I am forwarding it to the ASM production staff for publication. Your paper will first be checked to make sure all elements meet the technical requirements. ASM staff will contact you if anything needs to be revised before copyediting and production can begin. Otherwise, you will be notified when your proofs are ready to be viewed.

Sincerely,
Brian Conlon
Editor
Microbiology Spectrum

Reviewer #1 (Comments for the Author):

My comments have been adequately addressed by the authors.

Reviewer #2 (Comments for the Author):

Thank you for your resubmission of the manuscript. After careful review, I am pleased to note that you have addressed the previously raised concerns comprehensively. The revisions have significantly improved the quality and clarity of the paper. Below are my final comments and minor suggestions to further refine your manuscript:

Clarity and Readability:

The overall structure and flow of the manuscript have improved. The introduction now provides a clearer context for the study, and the objectives are well-defined.

Consider revising a few sentences in the conclusion to enhance readability and ensure the main findings are succinctly summarized.

Literature Review:

The expanded literature review is comprehensive and provides a solid background for your research. Ensure that all references are up-to-date and relevant to the topic.

Methodology:

The methodology section is now more detailed and transparent. This adds to the reproducibility of the study. Make sure that all methodological steps are clearly outlined for the benefit of readers who may wish to replicate the study.

Data Analysis:

The data analysis section has been improved with clearer explanations and justifications for the statistical methods used. Double-check all statistical data and calculations to maintain accuracy.

Results and Discussion:

The results are well-presented and discussed in the context of the existing literature. Ensure that all figures and tables are correctly labeled and referenced in the text.

The discussion now appropriately highlights the significance of the findings and their implications for the field. Consider adding a few sentences about potential limitations and future research directions.

Technical Aspects:

The manuscript is generally well-written, with few typographical errors. A final proofread is recommended to eliminate any remaining minor errors.

Ensure that the formatting adheres to the journal's guidelines, particularly concerning headings, subheadings, and reference styles.

Overall, the manuscript is now of high quality and contributes valuable insights to the field. I commend you for your efforts in addressing the feedback and making substantial improvements. I am pleased to recommend this revised manuscript for publication.

Best regards,

Dr Hayat